# Autonomous mobile robots for exploratory synthetic chemistry

Tianwei Dai[1,2], Sriram Vijayakrishnan[1,2], Filip T. Szczypiński[1,2], Jean-François Ayme[1,2], Ehsan Simaei[1,2], Thomas Fellowes[1], Rob Clowes[1], Lyubomir Kotopanov[1], Caitlin E. Shields[1], Zhengxue Zhou[1], John W. Ward[1] & Andrew I. Cooper[1✉]

Autonomous laboratories can accelerate discoveries in chemical synthesis, but this requires automated measurements coupled with reliable decision-making[1,2]. Most autonomous laboratories involve bespoke automated equipment[3–6], and reaction outcomes are often assessed using a single, hard-wired characterization technique[7]. Any decision-making algorithms[8] must then operate using this narrow range of characterization data[9,10]. By contrast, manual experiments tend to draw on a wider range of instruments to characterize reaction products, and decisions are rarely taken based on one measurement alone. Here we show that a synthesis laboratory can be integrated into an autonomous laboratory by using mobile robots[11–13] that operate equipment and make decisions in a human-like way. Our modular workflow combines mobile robots, an automated synthesis platform, a liquid chromatography–mass spectrometer and a benchtop nuclear magnetic resonance spectrometer. This allows robots to share existing laboratory equipment with human researchers without monopolizing it or requiring extensive redesign. A heuristic decision-maker processes the orthogonal measurement data, selecting successful reactions to take forward and automatically checking the reproducibility of any screening hits. We exemplify this approach in the three areas of structural diversification chemistry, supramolecular host–guest chemistry and photochemical synthesis. This strategy is particularly suited to exploratory chemistry that can yield multiple potential products, as for supramolecular assemblies, where we also extend the method to an autonomous function assay by evaluating host–guest binding properties.

Autonomous robotic laboratories have the potential to change our approach to chemical synthesis, but there are barriers to their widescale adoption. Autonomy implies more than automation; it requires agents, algorithms or artificial intelligence to record and interpret analytical data and to make decisions based on them[14,15]. This is the key distinction between automated experiments, where the researchers make the decisions, and autonomous experiments, where this is done by machines. The efficacy of autonomous experiments hinges on both the quality and the diversity of the analytical data inputs and their subsequent autonomous interpretation. Automating the decision-making steps in exploratory synthesis[16] is challenging because, unlike some areas of catalysis[11], it rarely involves the measurement and maximization of a single figure of merit. For example, supramolecular syntheses can produce a wide range of possible self-assembled reaction products[17], presenting a more open-ended problem from an automation perspective than maximizing the yield of a single, known target. Exploratory synthesis lends itself less well to closed-loop optimization strategies, at least in the absence of a simple quantitative 'novelty' or 'importance' metric.

In manual exploratory synthesis, reactions are usually characterized by more than one technique to allow the unambiguous identification of the chemical species. For example, in small-molecule organic syntheses and supramolecular chemistry, mass spectrometry (MS) and nuclear magnetic resonance (NMR) spectroscopy are often combined to probe molecular weight and molecular structure, respectively. Automating the analysis of such multimodal analytical data to guide synthetic discovery processes is not trivial[18]. Artificial-intelligence-based approaches, confined by their training data, might impede genuinely new discoveries by adhering too closely to established prior knowledge. Likewise, rule-based decision methods require careful implementation lest they overlook chemistry that deviates from the rules. More fundamentally, synthetic diversity leads to diverse characterization data. For example, some products in a library might yield highly complex NMR spectra but simple mass spectra, whereas other compounds may show the reverse behaviour, or perhaps give no mass signals at all. As chemists, we make routine, context-based decisions about which data streams to focus on, but this is a major hurdle for autonomous systems.

Much progress has been made towards diversifying automated synthesis platforms[4,5,19] and increasing their autonomous capabilities[9,14,15,20–22]. So far, most platforms use bespoke engineering and physically integrated analytical equipment[6]. The associated cost,

[1]Leverhulme Research Centre for Functional Materials Design and Materials Innovation Factory, University of Liverpool, Liverpool, UK. [2]These authors contributed equally: Tianwei Dai, Sriram Vijayakrishnan, Filip T. Szczypiński, Jean-François Ayme, Ehsan Simaei. ✉e-mail: aicooper@liverpool.ac.uk

complexity and proximal monopolization of analytical equipment means that single, fixed characterization techniques are often favoured in automated workflows, rather than drawing on the wider array of analytical techniques available in most synthetic laboratories. This forces any decision-making algorithms to operate with limited analytical information, unlike more multifaceted manual approaches. Hence, closed-loop autonomous chemical synthesis often bears little resemblance to human experimentation, either in the laboratory infrastructure required or in the decision-making steps.

We showed previously[11] that free-roaming mobile robots could be integrated into existing laboratories to perform experiments by emulating the physical operations of human scientists. However, that first workflow was limited to one specific type of chemistry—photochemical hydrogen evolution—and the only measurement available was gas chromatography, which gives a simple scalar output. Subsequent studies involving mobile robots also focused on the optimization of catalyst performance[12,13]. These benchtop catalysis workflows[11–13] cannot carry out more general synthetic chemistry, for example, involving organic solvents, nor can they measure and interpret more complex characterization data, such as NMR spectra. The algorithmic decision-making was limited to maximizing catalyst performance[11], which is analogous to autonomous synthesis platforms that maximize yield for a reaction using NMR[23] or chromatographic[10,24] peak areas.

Here we present a modular autonomous platform for general exploratory synthetic chemistry. It uses mobile robots to operate a Chemspeed ISynth synthesis platform, an ultrahigh-performance liquid chromatography–mass spectrometer (UPLC-MS) and a benchtop NMR spectrometer. This modular laboratory workflow is inherently expandable to include other equipment, as shown here by the addition of a standard commercial photoreactor.

To tackle a broad range of chemistry targets, a heuristic decision-maker was developed to process orthogonal NMR and UPLC-MS data, thus autonomously selecting successful reactions for further study without any human input. This decision-maker also checked the reproducibility of any hits from reaction screens before scale-up. This synthesis–analysis–decision cycle mimics human protocols to make autonomous decisions on the subsequent workflow steps. We exemplify the approach through structural diversification chemistry and the autonomous identification of supramolecular host–guest assemblies. Although the syntheses were autonomous, the choice of chemistry was not: the reactions and building blocks were selected by domain experts before the experiments. This nonetheless gave a large reaction space for the decision-maker to navigate. We also extended this autonomous approach beyond synthesis to assay function by autonomously assessing the host–guest binding properties of successful supramolecular syntheses.

## Modular robotic workflow

We partitioned our platform into physically separated synthesis and analysis modules (Fig. 1a,b). The physical linkage between the modules was achieved by using mobile robots for sample transportation and handling (Supplementary Videos 1–5). As these robots are free-roaming, the relevant instruments can be located anywhere in the laboratory. There is no limit to the number of instruments that can be incorporated under this paradigm, other than those imposed by laboratory space.

We chose a commercial Chemspeed ISynth synthesizer as the synthesis module, but the approach should be transferable to any automated synthesis platform that requires ex situ analyses. ISynth synthesizers have some built-in capability for integrating certain analytical equipment[24], but we chose here to integrate the analytical devices in a more distributed, scalable way.

Reactions were monitored by UPLC, MS and NMR to achieve a characterization standard comparable to manual experimentation.

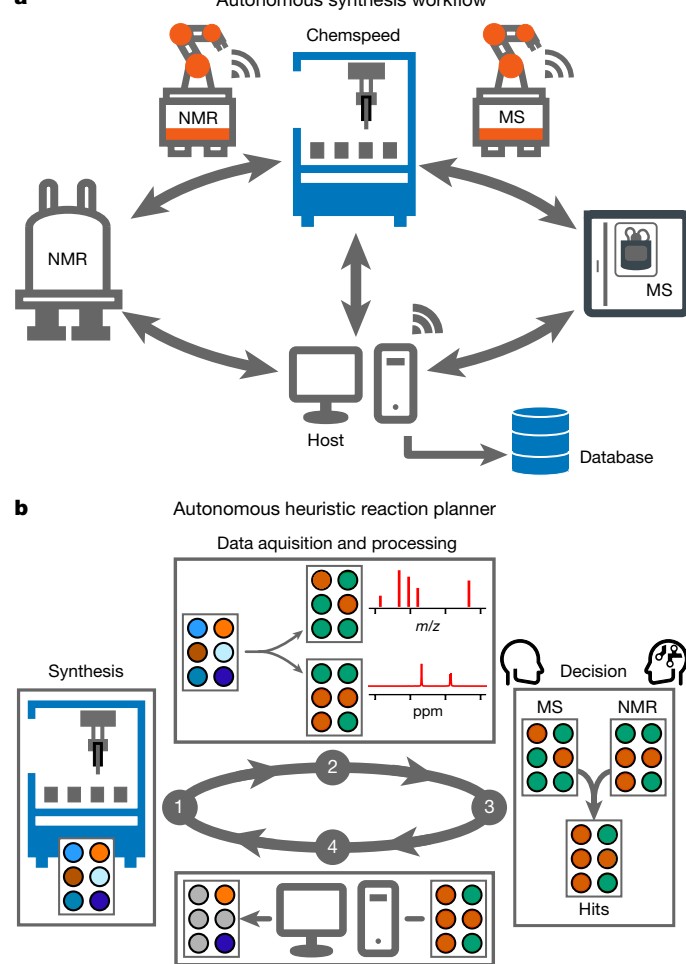

**Fig. 1 | Modular robotic workflow and heuristic reaction planner. a**, Workflow design for autonomous synthesis using mobile robots for sample transport and handling between the ISynth synthesizer and the MS and NMR analysis stations. **b**, An application-agnostic heuristic reaction planner takes input from the MS and NMR measurements and combines these two orthogonal datasets to plan the next stage in the synthesis experiment. Only reactions that pass predetermined, application-specific thresholds for both the MS and the NMR measurements are deemed to be hits, which are then taken forward for scale-up or for further reproducibility tests (green, pass; orange, fail).

The combination of orthogonal analytical techniques is essential to capture the diversity inherent in modern organic chemistry[25] and to mitigate the uncertainty associated with relying solely on unidimensional measurements[18]. To do this, our platform involves the periodic use of remotely located, unmodified 80-MHz benchtop NMR and UPLC-MS instruments. This set-up allows the use of standard laboratory consumables for these instruments and, importantly, allows the instruments to be shared with other automated workflows or used by human researchers in between measurements.

On completion of a chemical synthesis, the ISynth synthesizer takes an aliquot of each reaction mixture and reformats it separately for MS and NMR analysis. Mobile robots are then used to handle the samples and to transport them to the appropriate instrument (Supplementary Videos 1–3). Electric actuators were installed on the ISynth door to allow automated access by the two robot agents. Other than this, the various instruments were physically unmodified. Data acquisition occurs autonomously after sample delivery by the mobile robots using a set of customizable Python scripts. The resulting data are saved in a central database (Fig. 1a).

At the end of a synthesis–analysis cycle, the data are processed by the algorithmic decision-maker, which determines the next synthesis operations based on a set of heuristics designed by scientists with domain expertise in the specific research area (Fig. 1b). In the workflows presented here, the decision-maker first gives a binary pass or fail grading to the MS and ¹H NMR analysis of each reaction, based on experiment-specific criteria that are determined by the domain expert. The binary results of each analysis are combined to give a pairwise, binary grading for each reaction in the batch, and the decision-maker then instructs the ISynth platform which experiments to perform next.

Previous examples of autonomous experimentation have often used chemistry-blind optimization approaches[11], but such methods encounter challenges when dealing with reactions whose outcomes, unlike catalyst activity or yield of a known product, are not unique and scalar. For example, supramolecular self-assembly processes can produce many different possible combinations from the same starting materials, frequently giving complex product mixtures. To address this, we designed a 'loose' heuristic decision-maker that remains open to novelty and, hence, to chemical discovery. This application-agnostic decision-maker is broad enough to be applied to any chemistry that involves characterization by UPLC-MS and ¹H NMR, within the limits of those two techniques, and sufficiently customizable enough to allow expert chemists to define experiment-specific pass or fail criteria. For example, in the experiments described here, reactions must pass both orthogonal analyses to proceed to the next step (Fig. 1b). The two analyses were considered equally important here, but it would also be possible to weight the importance of each analytical method, for example, to favour either the MS or the NMR results.

The entire platform is operated through control software on a host computer that orchestrates the specified workflow. This control software also allows the development of analytical and synthesis routines by domain experts without any background in robotics.

Initially, we chose to use two task-specific mobile robotic agents to demonstrate the scalability of our approach into large industrial labs where a single robot would not provide sufficient capacity. However, this led to significant equipment redundancy. We therefore also demonstrated that the workflow tasks can be performed using a single mobile robot fitted with a multipurpose gripper (Supplementary Video 5 and Extended Data Fig. 1).

## Parallel synthesis for structural diversity

The synthesis of libraries is a bottleneck in the design–make–test–analyse cycle for drug discovery, and parallel synthesis has been used routinely to address this[26]. A typical synthetic workflow might involve the attempted synthesis of several common precursor molecules, followed by the scale-up of successful substrates to be further elaborated in a divergent synthesis. The decision to take a given reaction forward to the next step is usually made by a researcher, both for manual reactions and for reaction libraries produced using automated synthesis platforms.

We sought to emulate this end-to-end process in our automated synthesis platform by performing an autonomous divergent multi-step synthesis involving reactions with medicinal chemistry relevance[27,28] (Fig. 2a), with no intermediate human interventions beyond chemical restocking, if required. First, the parallel synthesis of three ureas and three thioureas was attempted through the combinatorial condensation of three alkyne amines (1–3), with either an isothiocyanate (4) or an isocyanate (5). The reaction mixtures were then analysed by UPLC-MS and ¹H NMR to identify samples where product formation had occurred. Our decision-maker parsed both data streams and directed the ISynth platform to scale up screening reactions that it had deemed to be successful. The decision-maker then analysed the scaled-up reactions by UPLC-MS and ¹H NMR to ensure parity with the initial screening-scale reactions. Only if the decision-maker deemed this scale-up attempt as successful would it then take that reaction forward, directing the ISynth to elaborate those precursor molecules. This emulates human pharmaceutical workflows, where modest scalability is generally a prerequisite for next-stage diversification, even in the discovery phase.

Two orthogonal diversification strategies were explored by the decision-maker in parallel: the Sonogashira cross-coupling of an alkyne with 2-bromopyridine (12), and a copper-catalysed azide–alkyne cycloaddition (CuAAC) with the antiretroviral active agent zidovudine (18). In principle, 12 different structurally diverse analogues could be accessed by this workflow based on 7 commercially available precursors via 3 autonomous synthetic steps. Researchers set the general success criteria for the decision-maker at the beginning of the workflow: beyond that, there was no human input as to which samples were chosen for scale-up or for subsequent diversification. The same success criteria were applied to all substrates; that is, the criteria were not tuned on a per-reaction basis.

Emulating a synthetic medicinal chemist, the heuristic decision-maker answered two basic questions: first, whether a chemical change from the starting materials had occurred; and second, whether the mass of the anticipated condensation product was among the main peaks identified in the UPLC trace (Fig. 2b). Chemical change was identified by the decision-maker through ¹H NMR spectroscopy. To do this, spectra corresponding to the starting materials were combined and a distance metric between their sum and the spectrum of the reaction mixture was calculated using dynamic time warping. Analysis of the UPLC-MS spectra involved automated peak detection in the UPLC trace, and extraction of the mass spectra corresponding to these identified peaks. Finally, experimental mass-to-charge ($m/z$) values were matched against calculated values that were generated algorithmically according to the expected chemistry. Only samples that the decision-maker deemed to have satisfied both the ¹H NMR and the UPLC-MS conditions were subsequently scaled up beyond the screening stage. Scaled-up samples were analysed by the decision-maker using the same method, but now a smaller difference in the dynamic time-warping metric was used as a threshold to confirm their parity with the screening samples (Methods). Given the complexity of the final molecules post diversification and the presence of multiple non-deuterated solvents, coupled with the high spatial overlap of resonances on a low-field benchtop NMR spectrometer, the decision-maker was programmed to use only UPLC-MS traces to assess final product formation at the diversification stage and ¹H NMR spectra of the crude reactions were recorded for future reference, rather than to inform pass or fail decisions.

The system operated for almost four consecutive days under this autonomous decision-making paradigm. The only researcher interventions required were restocking of the chemicals needed for the scale-up and diversification steps that the decision-maker deemed successful, along with fresh consumables (Fig. 2b). In principle, those restocking steps could be automated, too, using mobile robots. Out of the six targeted (thio)ureas, five substrates (6–10) were identified by the decision-maker as being successful. The decision-maker therefore instructed the ISynth platform to perform a scale-up experiment on the selected five reactions, and then re-analysed the scaled-up data. All five attempted scale-up replications yielded the products expected from the screening trial, so the decision-maker elected to take all five reactions forward to attempt the Sonogashira and CuAAC diversification steps (Extended Data Fig. 2). Although target molecules 14 and 15 were successfully synthesized, 13 underwent an unexpected intramolecular cyclization reaction (Extended Data Fig. 3). This was identified by human inspection of the NMR data and confirmed through single-crystal X-ray diffraction measurements. This species has the same molecular weight as the uncyclized cross-coupling product and was not distinguishable by UPLC chromatograms or MS alone. This highlights both the need for orthogonal characterization methods and the limitations of autonomous decision-making for such unexpected edge cases. Overall, the CuAAC reactions produced four of the five

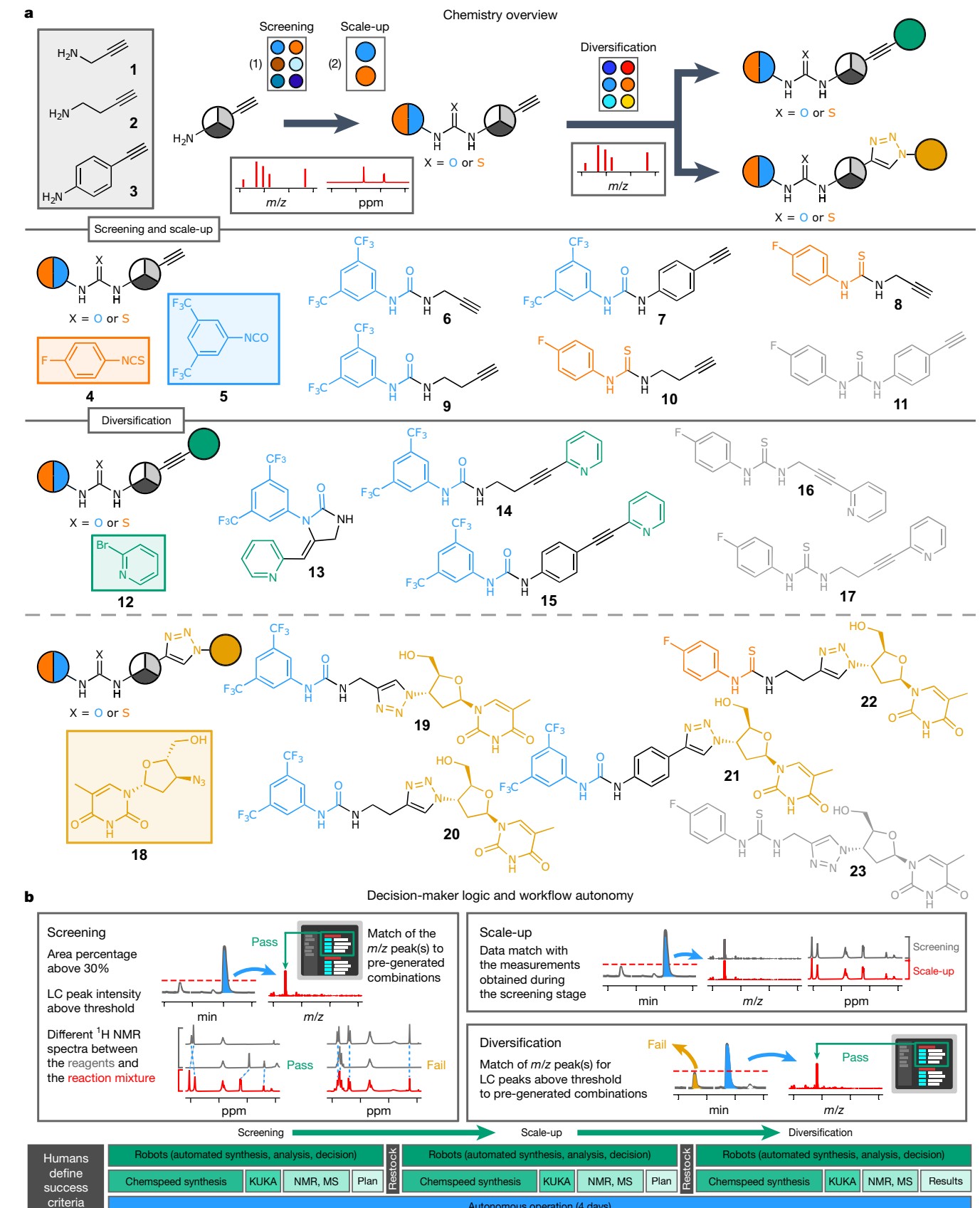

**Fig. 2 | Autonomous divergent syntheses. a**, Overview of the synthetic route for used for parallel synthesis. The coloured chemical structures were produced successfully, whereas the greyed-out structures were not, according to the rules of the decision-maker. **b**, Heuristic decision-maker logic for screening, scale-up replication and diversification steps. The timeline (bottom) shows tasks performed by humans in black boxes and tasks performed by the autonomous platform in green boxes. See Supplementary Information section 2, for reaction conditions. KUKA refers to the mobile robots.

target molecules (**19**–**22**). Final stage diversification reactions that were deemed by the decision-maker to be successful were purified offline by automated flash chromatography for full characterization of the product, as is commonplace in medicinal chemistry discovery programmes. Manual inspection of the data post-experiment confirmed that the decision-maker software had performed well, making essentially the same autonomous decisions that would have been made by a medicinal chemist in a manual workflow.

## Autonomous discovery of host–guest assemblies

Exploratory supramolecular synthesis presents different challenges for an autonomous robotic platform. The aim is to direct self-assembly processes to produce organized synthetic architectures with one practical motivation being the creation of molecular hosts[29,30]. However, supramolecular reactions can lead to a wide range of products, and often product mixtures. Even if a single supramolecular product is formed, host–guest binding properties can be remarkably sensitive to three-dimensional host structure and to solvent choice. Hence, the discovery of host–guest systems usually requires laborious trial-and-error experiments. Our autonomous synthetic chemistry platform lends itself perfectly to the acceleration of such discoveries. Unlike the medicinal chemistry example, above, where a predetermined product was the target, we set out here to 'expect the unexpected' by leveraging the combined use of coupled orthogonal analytical techniques. To do this, we did not limit the algorithm to follow the disappearance and appearance of specific signals, as in the divergent synthesis example, but instead directed our decision-maker to pursue any reaction from the screening stage that shows a $^1$H NMR spectrum that suggests a symmetric self-assembled architecture combined with an $m/z$ value in the MS that corresponds to any stoichiometrically reasonable metal–organic assembly. These allowed stoichiometric $m/z$ combinations were calculated beforehand based on the molar masses of the constituent building blocks and metal–ligand bonding rules. Structures that the decision-maker deemed to have passed the screening stage were then autonomously tested for reproducibility by the decision-maker instructing the ISynth platform perform six replicate reactions. Here we also extended the workflow to property screening, and any supramolecular hosts formed by reproducible syntheses were tested by the decision-maker autonomously for function by qualitative assessment of guest encapsulation, with the decision-maker finally determining which guests were bound. As in the parallel synthesis workflow, human researchers set only the general success criteria before the workflow was started and did not subsequently intervene as to what supramolecular species the decision-maker would choose to take forward for replication or for host–guest binding studies.

The autonomous experiment ran continuously over 3 days with the objective of discovering supramolecular architectures that can bind a small library of six structurally related guest molecules (Fig. 3a). In the first screening step, we explored the combinatorial condensation of three carbonyl-containing pyridines, **24**–**26**, and three bidentate and tridentate amines, **27**–**29**, in the presence of a metal ion (Cu$^+$ or Zn$^{2+}$). The reaction mixtures were analysed by direct-injection MS and by $^1$H NMR to identify samples that might contain promising candidate supramolecular hosts. Before the experiment, we calculated the molar masses for all possible chemically meaningful stoichiometric imine-based metal complexes assuming maximal site occupancy of the metal cations (up to 10 or 12 metal cations for Zn$^{2+}$ or Cu$^+$, respectively). For each of these complexes, we then predicted the $m/z$ values corresponding to any possible number of counterions present (Fig. 3b). To be deemed interesting, we set a general rule that the mass spectrum needed to contain at least two different charge combinations for any of the possible precalculated metal–organic architectures. This rule was established to avoid the false positives

that might occur for such a large $m/z$ look-up table, which had 110 separate entries (1,110 $m/z$ values in total).

Instead of directing the decision-maker to focus on the disappearance of the $^1$H NMR signals for the starting materials[25], it was programmed to look for samples that contained a number of NMR peaks that was comparable to the sum of the number of peaks in the starting material spectra, but with different chemical shifts. This approach maximizes the chances of identifying reaction mixtures that yield a single symmetric architecture, but it would deselect, for example, complex product mixtures with large numbers of NMR signals, such as coordination polymers or oligomers, even in cases where the starting materials are consumed. Because we chose criteria that were loose enough to encourage serendipitous discovery, each of the two characterization tests yielded false positives when taken in isolation. However, as with the divergent syntheses, above, we informed the decision-maker to require both $^1$H NMR and MS criteria to be passed independently to proceed to the replication step, thus minimizing such false positives. All spectra, including reactions that were deemed to have failed, were saved in our database and were available for manual inspection. One technical limitation was that the high dispersion related to the low field strength of the benchtop NMR instrument (80 MHz) can lead to an artificial increase in the apparent number of peaks. Again, this limitation was offset by using two orthogonal characterization methods—in this, the requirement to match plausible $m/z$ ratios in the MS measurement.

Using these search criteria, 2 out of the 18 possible combinations of carbonyl-bearing pyridines with amines and metals were deemed successful by the decision-maker and carried forward to the subsequent experimental stages (Extended Data Fig. 4): these corresponded to a known metal–organic cage[31] [Zn$_4$(**24**$_3$,**28**)$_4$]$^{8+}$ and a metal–organic helicate[32] [Zn$_2$(**24**$_2$,**29**)$_3$]$^{4+}$. The decision-maker then instructed the ISynth platform to replicate both of these reactions six times each, and checked these replicates for parity with $^1$H NMR and MS measurements from the initial screening stage. Having established repeatability, the decision-maker then instructed the ISynth to proceed with guest binding studies, in which aliquots of six small organic molecules were dispensed into the six replicated candidate host solutions. The mixtures were then subjected to $^1$H NMR analysis to identify binding-induced changes in the spectra. Some of the example guests were expected to exchange slowly with the host on the NMR timescale. To emulate the effects of fast NMR exchange and to simplify the analysis, a large line-broadening value was applied to the collected data. Guests were identified qualitatively as being 'bound' if the decision-maker algorithm identified a change to the chemical shift in the aromatic region of $^1$H NMR upon guest addition. Three guests were found by the decision-maker to successfully bind inside the cavity of the cage [Zn$_4$(**24**$_3$,**28**)$_4$]$^{8+}$ whereas no guest interacted with the helicate [Zn$_2$(**24**$_2$,**29**)$_3$]$^{4+}$ (Extended Data Fig. 5), in agreement with related studies[30,31] that show that helicates lack a host binding cavity.

## Offline photochemical synthesis

Our automated platform was designed to be modular and flexible, and it is easy to integrate other physical modules into the workflow. For example, our ISynth platform does not have a photochemical reaction capability, but we solved this by appending a commercial, standalone photoreactor as an additional remote module (Fig. 4a and Supplementary Video 4).

We showcased this with a catalyst-screening experiment for decarboxylative conjugate addition[33] of the protected amino acid N-(*tert*-butoxycarbonyl)-proline to diethyl benzylidenemalonate[34] (Fig. 4b). Following inertization of the ISynth platform and dispensing of the liquid reagents and solvents, the samples were crimp-sealed under a nitrogen atmosphere and transported by one of the robot agents to the offline photoreactor station for irradiation. The samples were then returned to the ISynth and reformatted for the analysis by UPLC–MS.

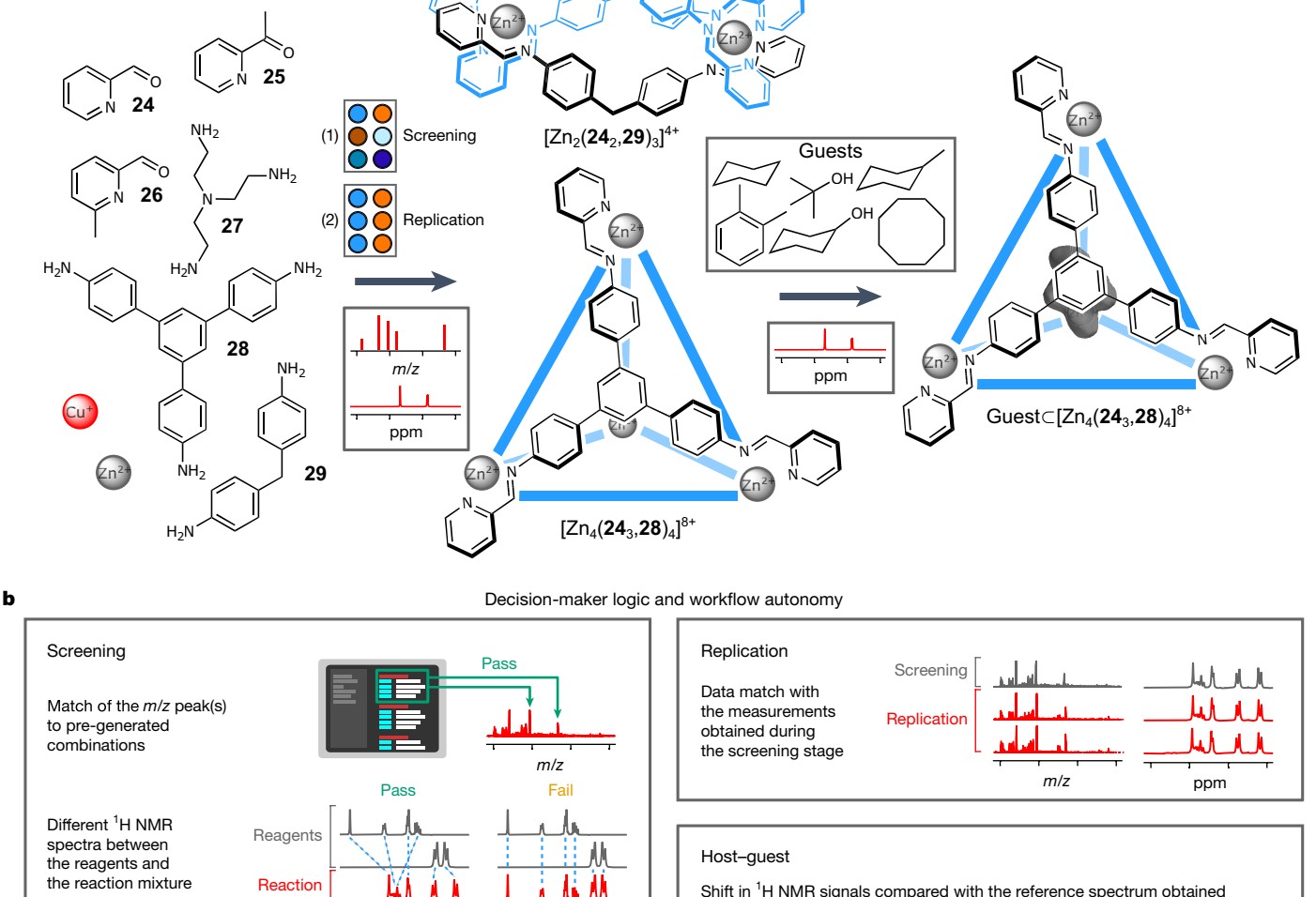

**Fig. 3 | Autonomous discovery of supramolecular host–guest systems.**
**a**, Combinations of amines, carbonyl-bearing pyridines, metal ions and guests used for autonomous supramolecular host–guest cage syntheses. **b**, Heuristic decision-maker logic used for screening, scale-out replication and host–guest binding experiments. In this case, the number of [1]H NMR peaks relative to the

starting materials was used as a threshold criterion. The timeline (bottom) shows tasks performed by humans in black boxes and tasks performed by the autonomous platform in green boxes. See Supplementary Information section 3, for reaction conditions.

In this example, there was substantial overlap of product resonances in the low-field benchtop [1]H NMR spectra that stemmed from the presence of diastereomers and rotamers, as well as the use of two non-deuterated solvents; hence, the decision-maker was programmed to assess final product formation using UPLC–MS traces alone (Fig. 4c). Three catalysts, 2,4,5,6-tetrakis(9H-carbazol-9-yl) isophthalonitrile (4CzIPN), [Ir(dtbbpy)(ppy)$_2$]PF$_6$ (where dtbbpy is 4,4′-di-*tert*-butyl-2,2′-bipyridine and ppy is (2-pyridinyl)phenyl)), and (Ir[dF(CF$_3$)ppy]$_2$(dtbbpy)) PF$_6$ (where dF(CF$_3$)ppy is 3,5-difluoro-2-[5-(trifluoromethyl)-2-pyridinyl] phenyl), were found to yield the desired decarboxylative conjugate

addition product. The other three photocatalysts (eosin Y, graphitic carbon nitride and 2,4,6-triphenylpyrylium tetrafluoroborate) and a blank control produced only starting materials, with no trace of the product.

## Conclusion

We have created a strategy for exploratory synthetic chemistry using mobile robots to integrate distributed synthesis and analysis platforms. Although these workflows are not closed loop, in that they

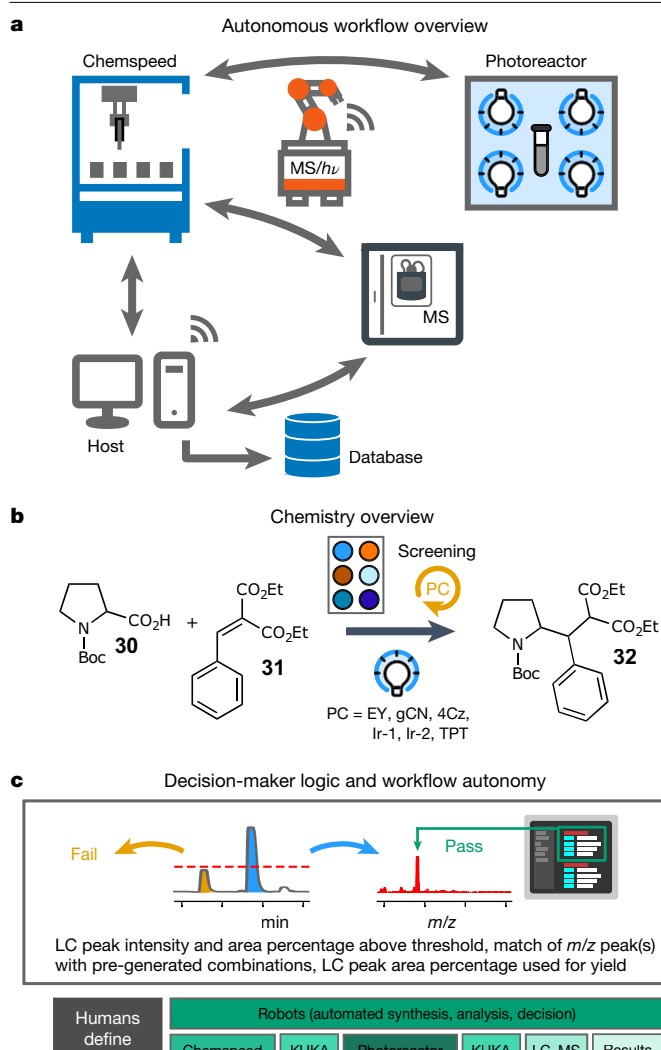

**a** Autonomous workflow overview

Chemspeed — Photoreactor — MS/hν — MS — Host — Database

**b** Chemistry overview

Screening — PC

PC = EY, gCN, 4Cz, Ir-1, Ir-2, TPT

**c** Decision-maker logic and workflow autonomy

Fail — Pass

LC peak intensity and area percentage above threshold, match of *m/z* peak(s) with pre-generated combinations, LC peak area percentage used for yield

Humans define success criteria — Robots (automated synthesis, analysis, decision) — Chemspeed | KUKA | Photoreactor | KUKA | LC–MS | Results — Autonomous operation (2 days)

**Fig. 4 | Addition of a photoreaction station to the modular, distributed workflow. a**, Workflow design for integration an offline photoreactor station into the synthesis workflow. **b**, Autonomous screening of photocatalysts for decarboxylative conjugate addition. **c**, Decision-maker logic for autonomous LC–MS data analysis. The timeline (bottom) shows tasks performed by humans in black boxes and tasks performed by the autonomous platform in green boxes. EY, eosin Y; gCN, graphitic carbon nitride; 4Cz, 4CzIPN; Ir-1, [Ir(dtbbpy) (ppy)$_2$]PF$_6$; Ir-2, (Ir[dF(CF$_3$)ppy]$_2$(dtbpy))PF$_6$; TPT, 2,4,6-triphenylpyrylium tetrafluoroborate. Boc, *N*-(*tert*-butoxycarbonyl). See Supplementary Information section 4, for reaction conditions.

of this structure remains undetermined, and manual attempts to produce crystals suitable for X-ray diffraction have so far failed. This illustrates how rule-based autonomous robotic searches can yield systems of potential interest, but also the challenge of characterizing them, even by hand. The inherent challenges associated with assessing the contextual novelty or importance of reactions in an autonomous way might suggest that we should focus instead on autonomous optimization of measurable function, such as in catalyst development[11], but not all areas of chemistry are function led, synthetic methodology development being one example.

These workflows could not have been performed with our earlier mobile robotic workflow[11] because they involve multi-step liquid additions with hazardous reagents and organic solvents, as well as characterization techniques that yield data streams that are significantly more complex. Our approach can tackle different types chemistry: for example, supramolecular reactions often yield complex product mixtures, whereas pharmaceutical diversification chemistry is designed, broadly speaking, to employ more predictable reaction steps.

This tiered heuristic decision-maker should be applicable to other synthetic chemistry problems where researchers would judge the combined outputs of UPLC–MS and [1]H NMR analyses, or by extension, other characterization methods. Such chemistry-specific heuristic approaches present an alternative to black-box machine learning models, and they might be better suited to analytically complex but data-sparse problems in synthetic organic chemistry. This heuristic implementation also captures expert human knowledge—for example, by pre-determining possible metal–organic stoichiometries using our knowledge of valency (Fig. 3)—which provides a focus for autonomous experimental searches in multi-dimensional chemical spaces that would otherwise be too large and too complex to navigate. Of course, such pre-programmed rules also introduce confirmation biases, and might miss important reactions, but the algorithmic workflow is fully traceable[4] and the data for all reactions, including 'unsuccessful' ones, are saved for future inspection. This mitigates the risk of missing potentially interesting, anomalous reactions that fail the predetermined decision thresholds (for example, Supramolecular Screening 9, above).

This modular approach should be scalable into the largest industrial laboratories, if necessary connecting physically separated synthesis and analytical labs using mobile robots that can traverse buildings[35]. In such a distributed scenario, the cost of the mobile robots might be a relatively minor consideration because industrial mobile robots, although still unfamiliar in laboratories, are a high-growth, highly commoditized technology serving multiple sectors beyond chemistry[36]. One practical learning point for modular workflows comprising multiple concatenated software and hardware platforms is the need for very low failure rates per module. For example, the workflow illustrated here required more than a year of development and debugging before it was stable enough to carry out these experiments.

Although we used a benchtop NMR here, the introduction of high-field automated NMR[37] might be necessary to characterize larger, more complex pharmaceutical molecules. Other future directions might include the development of more advanced algorithms for closed-loop synthesis optimization, conceivably folding in autonomous insights drawn from existing literature, or the implementation large language models[38,39] as an interface to improve accessibility for researchers inexperienced in automation.

## Online content

are not optimization processes, they do nonetheless involve autonomous decision-making steps that accelerate discoveries. The level of autonomous decision-making and contextual understanding is, of course, far lower than for a human researcher (Fig. 1b), but the system out-performs humans in other ways. For example, the algorithmic decisions are effectively instantaneous, providing a large acceleration over human workflows where a researcher would need to inspect all the characterization data before proceeding further. These autonomous searches led to new chemical understanding, although that did require additional post-experiment analysis by human researchers, for example, to identify the unexpected cyclization product shown in Extended Data Fig. 3. Likewise, in the supramolecular workflow, Supramolecular Screening 9 gave an [1]H NMR spectrum that passed the algorithm's hit threshold but failed the UPLC–MS test (no matching ions observed; Supplementary Scheme 32 and Supplementary Fig. 121). The full nature

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

## Methods

### Robot specifications

The KUKA robot agents have the native capability to navigate the laboratory environment and are equipped with an array of technologies such as motor encoders, inertial measurement units, laser scanners and force sensors. These agents are cobots; that is, they are commercially certified to be used in environments with humans without additional containment. The robots can determine their position with an accuracy of ±0.12 mm over any distance of travel, with an orientation precision of $\theta \pm 0.005°$. This level of precision is sufficient for all the manipulations in these synthetic workflows, and there were no positioning or placement errors in any of the experiments described here.

### Host control software

We developed an 'Intelligent Automation System Control Panel' (IAS-CP) to control the automation workflow in a modular and flexible way. Instrument-control computers were connected to the IAS-CP host PC using ZeroMQ broadcast and subscription capabilities. During workflow execution, the host PC broadcasts individual commands or sends pre-defined automated sub-workflows to the subscribers. Execution of specific instructions on the instrument computers via separate drivers is triggered upon receiving these commands (see 'Code availability' for instrument-control drivers). A subsequent workflow command is broadcast after the host receives a notification of successful task completion. A simple custom-made graphical user interface enables non-expert users to orchestrate and manage scheduling of the automation process. This modular design allows for development of client-side code for the device to integrate equipment with the scheduler (see https://doi.org/10.5281/zenodo.11197259 for an example of integrating further equipment into the network).

### Experimental stations

Apart from the photocatalysis experiments, all synthesis experiments were performed using a Chemspeed ISynth platform. The ISynth module enables a range of functions including liquid dispensing, heating, shaking, inertization, solvent evaporation and the parallel preparation of multiple samples. The platform was equipped with ISynth reactor blocks for heating and shaking, along with a crimp-capper module for microwave vial sealing under nitrogen atmosphere for offline photocatalysis experiments. The platform was modified with a pair of electric actuators to facilitate automated hood-door opening (Supplementary Fig. 192), along with a pair of light curtains to mitigate the risk to human and robotic operators.

### Liquid chromatography–mass spectrometry

The UPLC–MS measurements were performed using a Waters Acquity UPLC fitted with an SQ Detector 2 mass detector. The UPLC–MS machine was equipped with an Automation Portal attachment to allow for robotic placement of UPLC–MS samples. LC was performed on an Acquity UHPLC BEH C18 column with a gradient of 5–95% acetonitrile/water (0.1% formic acid) over 2 min, and held for 0.5 min with a further 1 min for re-equilibration and gradient reversal. Apart from the supramolecular chemistry workflow, blank injections of acetonitrile were performed between samples to ensure that no residual material remained on the column. For direct-injection measurements, the C18 column was bypassed, and acetonitrile (no formic acid) was used as the eluent. Samples were submitted to the AutoLynx software through a generated CSV file, which triggered automatic start-up of the machine from standby. The instrument returns to standby mode several minutes after the last sample in the batch is completed.

### NMR measurements

The NMR spectra in robotic workflow runs were collected on a Bruker Fourier80 (80.13 MHz) benchtop instrument. The use of a benchtop NMR also alleviated the need for deuterated solvents, opting instead for solvent-suppression sequences. The standard shim sample was used for constant shimming using the quickshim method when the instrument was not in operation for the workflow. As the internal lock sample was used, non-deuterated solvents were used for all measurements. Solvent suppression was performed using the MULTISUPPDC parameter set in TopSpin 4.3.0 with the noesycpdcgpps1d pulse programme, which uses the nuclear Overhauser effect spectroscopy presat method and $^{13}C$ decoupling. For all workflow samples, 64 scans were used. Single-scan unsuppressed $^1H$ NMR spectra (zg30 pulse programme) acquired within the MULTISUPPDC method were used to established spectra reference frequencies. The data from the Fourier80 instrument were acquired and analysed using a custom software package (https://doi.org/10.5281/zenodo.11174257) communicating with TopSpin 4.3.0 through official TopSpin Python API distributed by Bruker. The dynamic time-warping procedure from ref. 40 was used to compare differences in NMR spectra.

Additional characterization NMR spectra were recorded on a Bruker Avance III 400 (400.13 MHz) or Avance III HD 500 (500.13 MHz) instrument with the zg30 pulse programme (for $^1H$ NMR) and zgpg30 (for $^{13}C$ NMR).

### Photocatalysis

Photocatalysis experiments were performed on SynLED parallel photoreactors with 465-nm irradiation. The photoreactors were controlled with a Raspberry Pi, which interfaced with the scheduler to trigger the on/off functions.

### Single-crystal X-ray diffraction measurements

Data were recorded using a Rigaku Synergy-DW diffractometer equipped with a HyPix Arc100 photon-counting detector. The temperature during data collection was controlled with an Oxford Cryosystems Cryostream 700 Plus cooling device. Data reduction, and absorption and other corrections, were performed using CrysAlisPro 1.171.44.46a (Rigaku Oxford Diffraction, 2024). Structures were solved within Olex2[41] by the intrinsic phasing method of SHELXT[42]. Heavy atoms were refined anisotropically using SHELXL[43] using full-matrix least squares minimization against $F^2$, and hydrogen atoms were placed geometrically and refined using a riding model. Hydrogen atoms bonded to nitrogen atoms were freely refined.

### UPLC–MS data processing

UPLC-MS data was parsed and analysed using custom software packages (https://doi.org/10.5281/zenodo.11174536 and https://doi.org/10.5281/zenodo.11174323) based on a software development kit provided by Waters through extraction of the raw coordinate data.

## Data availability

Raw data, example input files, expected decision-making output and walkthrough examples of the decision-making parts of the code for each workflow have been deposited on Zenodo under a CC-BY-SA 4.0 license at https://doi.org/10.5281/zenodo.11197259 (ref. 44). Crystal structure data have been deposited at the Cambridge Crystallographic Data Centre with the identifiers 2355749 and 2355750.

## Code availability

All code has been deposited on GitHub under the MIT License: the driver for controlling the Bruker Fourier80 NMR (https://doi.org/10.5281/zenodo.11174257), the Python package port of the Waters SDK (https://doi.org/10.5281/zenodo.11174323), the parser for the Waters UPLC-MS.RAW files (https://doi.org/10.5281/zenodo.11174536) and the decision-maker (https://doi.org/10.5281/zenodo.11209893).

40. Giorgino, T. Computing and visualizing dynamic time warping alignments in R: the dtw package. *J. Stat. Softw.* **31**, 1–24 (2009).

41. Dolomanov, O. V., Bourhis, L. J., Gildea, R. J., Howard, J. A. K. & Puschmann, H. OLEX2: a complete structure solution, refinement and analysis program. *J. Appl. Crystallogr.* **42**, 339–341 (2009).
42. Sheldrick, G. M. SHELXT—integrated space-group and crystal-structure determination. *Acta Crystallogr.* **71**, 3–8 (2015).
43. Sheldrick, G. M. Crystal structure refinement with SHELXL. *Acta Crystallogr. C* **71**, 3–8 (2015).
44. Ayme, J.-F., Cooper, A. I., Szczypiński, F. T. & Vijayakrishnan, S. Data and code examples for: Twin cooperative mobile robots for autonomous synthetic chemistry. *Zenodo* https://doi.org/10.5281/zenodo.11209807 (2024).

**Acknowledgements** We acknowledge funding from the Leverhulme Trust via the Leverhulme Research Centre for Functional Materials Design. This project has received funding from the European Research Council (ERC) under the European Union's Horizon 2020 research and innovation programme (grant agreement number 856405). We received funding from the Engineering and Physical Sciences Research Council (EPSRC, EP/T031263/1 and EP/N004884/1). A.I.C. thanks the Royal Society for a Research Professorship (RSRP\S2\232003). We thank H. Fakhruldeen and G. Pizzuto for discussions and advice on robotics aspects; P. Wang for assistance with 3D-printing of vial holders; and the University of Liverpool Chemistry workshop for fabrication of metal brackets, and for installation of actuators onto the Chemspeed platform. F.T.S. thanks the Bruker UK applications scientists R. Stein and M. Howard for assistance with tuning the NMR parameters and advice on TopSpin command line interface. We acknowledge high-resolution quadrupole time-of-flight MS measurements provided by S. Robinson and R. Roberts at University of Liverpool.

**Author contributions** A.I.C. supervised and directed the research. T.D. developed the control software, microcontrollers and lab communication network to operate the workflow; and programmed both mobile robot platforms. S.V., J.-F.A., F.T.S. and A.I.C. designed and planned the chemistry experiments. E.S. designed the custom rack for the NMR, the corresponding gripper and integrated the NMR-Agent into the workflow. S.V. and J.-F.A. performed the manual and automated chemistry experiments, and characterized the products with help from F.T.S. S.V., F.T.S. and J.-F.A. developed the drivers for the UPLC–MS and NMR, and implemented the decision-maker. T.F. performed the crystallography experiments. C.E.S. developed an initial integration of the custom NMR rack onto the Chemspeed platform. Z.Z. and L.K. assisted with the programming of NMR-Agent and UPLC-Agent. R.C. designed, built and installed custom hardware for the workflow. L.K. integrated the use of a single mobile robotic agent. J.W.W. supervised the photocatalysis work and helped conceive the associated automation workflow. S.V., F.T.S., J.-F.A. and A.I.C. prepared the paper with contributions from T.D. and E.S.

**Competing interests** The authors declare no competing interests.

**Additional information**
**Correspondence and requests for materials** should be addressed to Andrew I. Cooper.

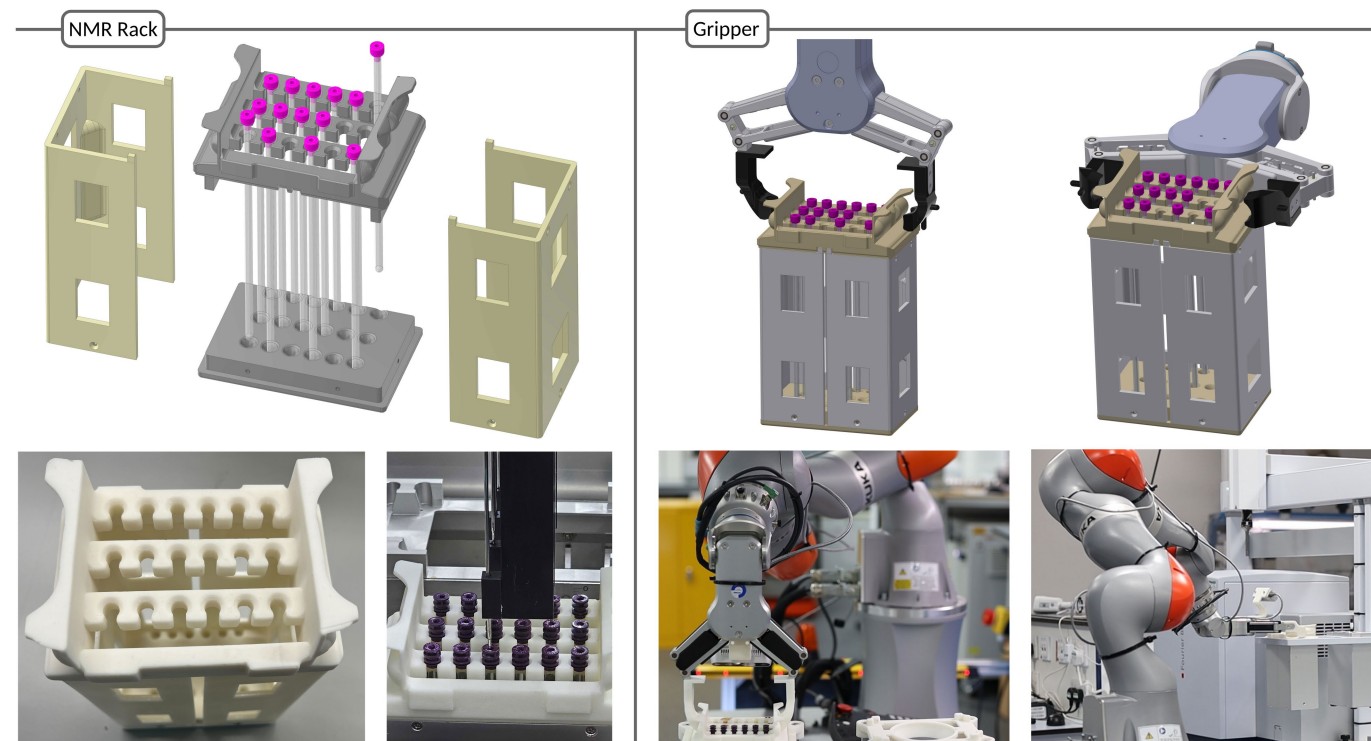

NMR Rack

Gripper

**Extended Data Fig. 1 | Custom-made NMR rack for mobile-agent handling. Left**. The assembled rack is pre-loaded with NMR tubes, and placed inside the ISynth platform; the liquid transfer tool then dispenses liquid into the tubes, which have caps with holes to allow for dispensing. **Right**. NMR-Agent uses custom-made fingertips that allow to it grasp and move the rack both vertically and horizontally. Vertical grip orientation is used to remove the rack from the ISynth deck, and horizontal grip orientation is used to move the rack into the benchtop NMR autosampler.

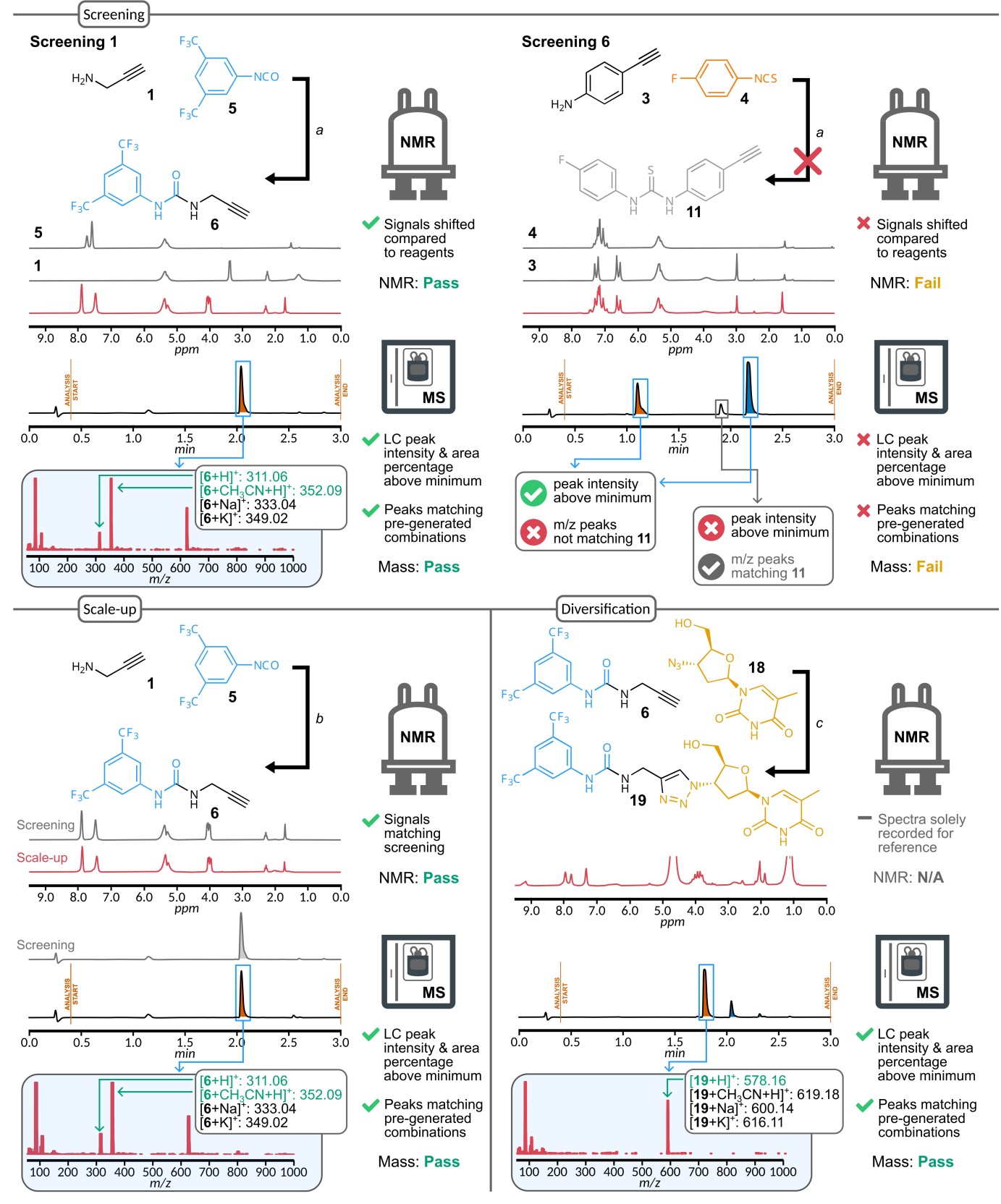

**Extended Data Fig. 2 | Decision maker rules for parallel divergent organic synthesis. Screening.** Attempted synthesis of urea **6** where NMR and LC-MS data both meet the respective criteria; urea **6** is hence deemed successful and taken forward for scale-up replication. For reaction target **11**, both NMR and LC-MS requirements failed and **11** is therefore deselected. **Scale-up.** NMR and LC-MS data from scale-up of synthesis of **6** is found to have parity with the screening sample and is judged as a 'pass'. **Diversification.** Attempted synthesis of CuAAc-catalyzed product **19** is verified by LC-MS to be successful. Conditions: **a.** DCM, r.t, 12 h; **b.** DCM, r.t, 12 h; **c.** Cu(SO$_4$), DCM:IPA:H$_2$O, ascorbic acid, N$_2$, 60 °C, 14 h.

**Extended Data Fig. 3 | Expected and observed structure for diversification synthesis.** Expected structure of **13**, as targeted in the parallel divergent organic synthesis diversification example (left) and the experimentally observed cyclized molecular structure of **13** and its single crystal x-ray structure (right).

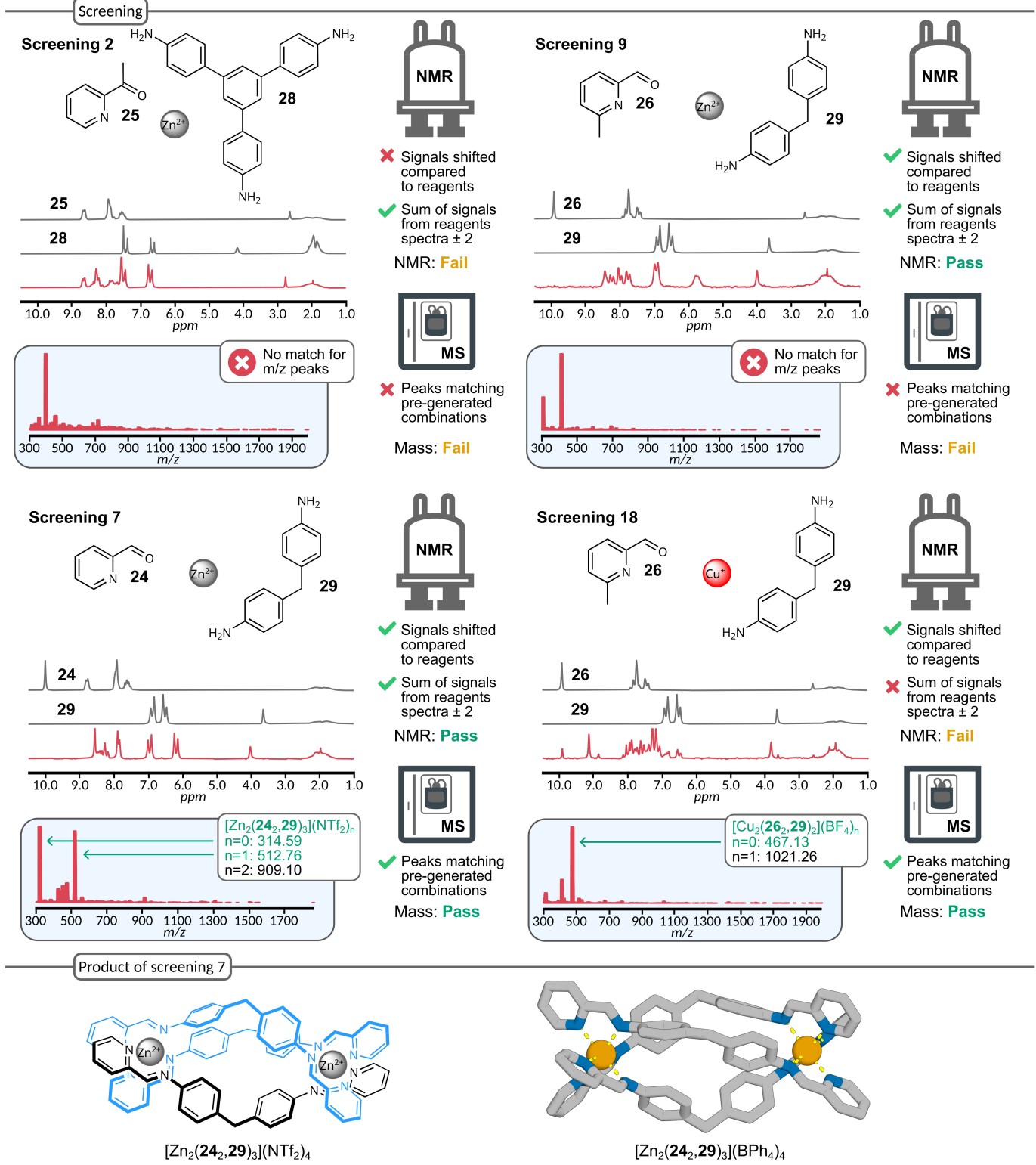

**Extended Data Fig. 4 | Decision maker rules for autonomous discovery of supramolecular host-guest assemblies. Screening.** Clockwise from top left. Screening data for sample 2 is judged by the decision maker as a failure, both by NMR and MS (red traces), because no MS hit is found, and the NMR signals arise largely from the starting materials; the reaction is therefore rejected for further replication. Screening sample 9 shows a similar number of [1]H NMR peaks to the starting materials but with substantial changes in position (*i.e.*, an NMR 'pass'), but there was no confirmation by MS, hence an overall 'fail'. Sample 18 shows changes in the NMR signals but the overall spectrum is complex and likely contains multiple species and the reaction is therefore rejected for replication, despite having a positive MS hit. Screening sample 7 passes all three NMR and MS criteria and it is selected for replication; the chemical structure and crystal structure of the resulting helicate are shown below.

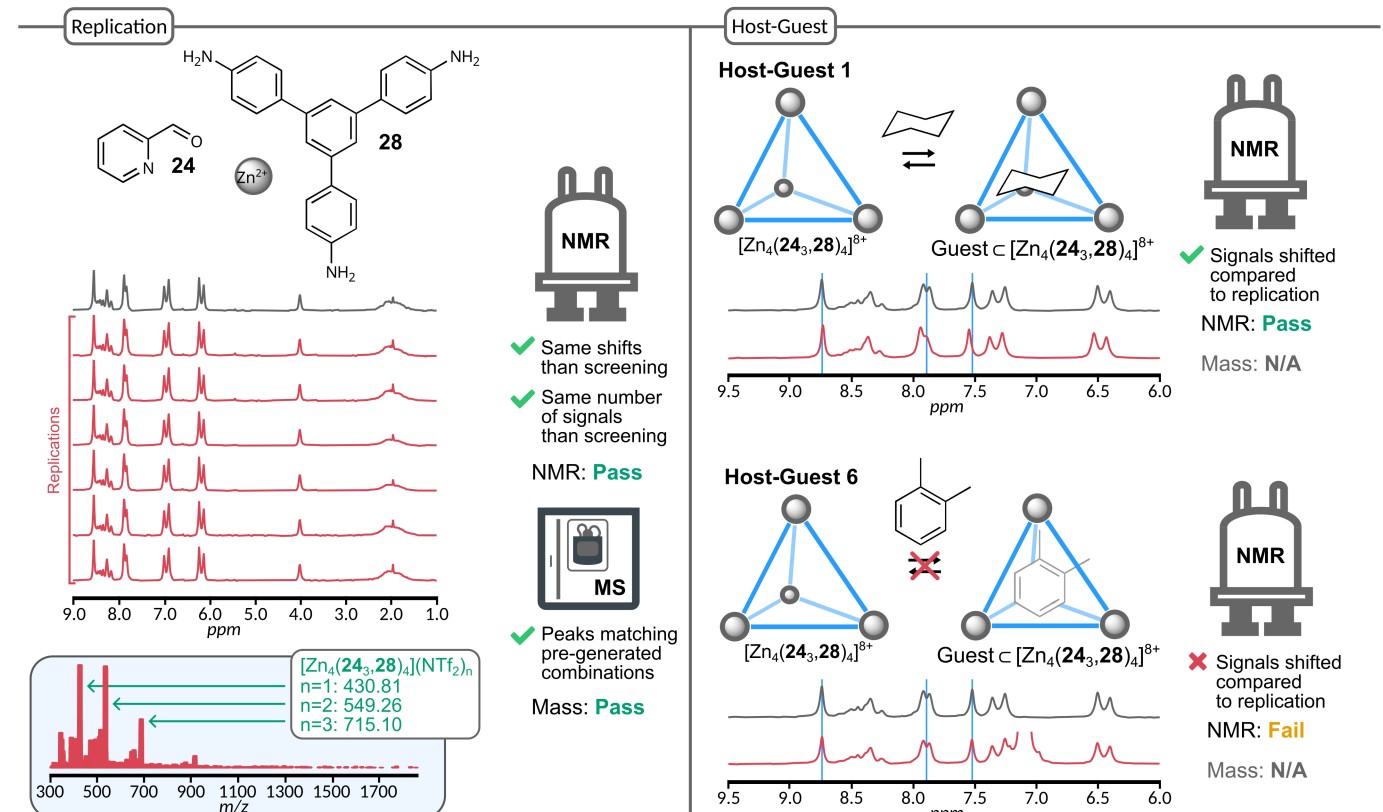

**Host-Guest 1**

$[Zn_4(24_3,28)_4]^{8+}$    Guest $\subset$ $[Zn_4(24_3,28)_4]^{8+}$

✔ Signals shifted compared to replication

NMR: **Pass**

Mass: **N/A**

**Host-Guest 6**

$[Zn_4(24_3,28)_4]^{8+}$    Guest $\subset$ $[Zn_4(24_3,28)_4]^{8+}$

✘ Signals shifted compared to replication

NMR: **Fail**

Mass: **N/A**

✔ Same shifts than screening

✔ Same number of signals than screening

NMR: **Pass**

✔ Peaks matching pre-generated combinations

Mass: **Pass**

$[Zn_4(24_3,28)_4](NTf_2)_n$
n=1: 430.81
n=2: 549.26
n=3: 715.10

**Extended Data Fig. 5 | Decision maker rules for autonomous discovery of supramolecular host-guest binding assemblies. Replication**. Data from the synthesis of a zinc cage in scale-out replication (red NMR traces) is checked for parity with NMR data from the screening stage (black traces) to ensure parity. Replicate samples are checked to pass both MS and NMR criteria, as for screening, to be judged as a 'pass'. The 6 autonomous NMR repeats shown here illustrate the reproducibility of the synthesis at this larger scale. **Host-Guest**. Binding of guests to the cage is validated by [1]H NMR. Cyclohexane successfully binds to the cage, as evidenced by shifts in the aromatic [1]H NMR signals, while xylene shows no such shifts and is deemed to be a host-guest binding 'fail'.