## [Peer Review File · Nature]

Manuscript Title: Autonomous mobile robots for exploratory synthetic chemistry

Reviewer Comments & Author Rebuttals

Reviewer Reports on the Initial Version:

Referees' comments:

Referee #1 (Remarks to the Author):

The manuscript by Cooper et al describes the development and use of an autonomous lab for chemistry that involves the integration of two mobile robots. This is following the team's seminal paper in 2020 in which they showed that a single mobile robot is capable of designing and performing photocatalysis experiments. In some ways, this manuscript is an advance over the prior work. For instance, they report three independent workflows that leverage two or more analytical techniques, namely NMR and mass spectrometry. Now that this automated lab has access to these tools along with a more sophisticated chemspeed platform for synthesis, it is capable of much more sophisticated and diverse chemistry. The paper does demonstrate this through the three different types of chemistry. That said, in other important ways, this work represents a step backwards from the prior work. Specifically, while the workflows and results of the work appear technically sound, they are less impressive and impactful than the prior work in three ways:

(1) The reliance on two mobile robots rather than one is a disadvantage. While having different systems working together could be interesting if they interacted in novel ways, the two robots just represent parallel shuttles that allow samples to be delivered to two places. Further, it seems that one robot could have done both jobs if the team had designed a sample holder that worked with both analytical instruments. The idea that a new robot must be incorporated to accommodate each new hardware module is not scalable and contrary to the notion that the advances in this work should be in principle applicable to other labs.

(2) The processes used to choose experiments seemed less sophisticated than prior work. It appears that the screening process simply comprised testing all available reactions. This would quickly become impractical for parameter spaces of reasonable size. The process of selecting which reactions to scale up had merit, but intelligence or decision making should be employed in selecting the experiments in the first place. Otherwise, it is not even clear this is a closed-loop process.

(3) In most papers describing the use of autonomous experimentation, the goal is to make better and smarter decisions than human researchers so that the system offers something beyond simply doing experiments faster. In this work, it appeared that the goal was to match the decision-making of humans. While it remains a virtue that they hit this target in terms of deciding which experiments to scale up, prior work has shown that autonomous systems can outperform human researchers, which makes the target set in this work a low bar.

These structural weaknesses aside, the team had a number of engineering achievements including

the use of analytical tools without modification and fully automated workflows with minimal human involvement. Unfortunately, these do not seem sufficient to justify publication of this work in a high profile journal rather than a targeted engineering journal.

Referee #2 (Remarks to the Author):

Summary

This manuscript describes using dual modes of chemical characterization to enable a heuristic reaction analyzer to come to a better-informed verdict on the outcome of a variety of reactions. Each mode of chemical characterization is handled by a dedicated mobile robot that can shuttle reaction samples from a ChemSpeed ISynth to the benchtop NMR and UPLC-MS, respectively. This dual characterization system was used to analyze the outcome of a series of drug-like molecule diversification reactions, host-guest interactions, guest assembly, and photocatalyst screening reactions. The strengths of the study are the demonstration of diverse applications and automation of outcome decisions based on expert rules, all using characterization equipment that is being shared concurrently with human users.

The advances seem incremental in view of the authors' previous contributions and out of step with recent reports of autonomous synthesis based on integrated generative and property machine learning models. The article title includes the word, "autonomous", but the manuscript does not describe to what extent autonomy is achieved. The authors state in the introduction that autonomy "requires agents, algorithms, or artificial intelligence (AI) to record and interpret analytical data and to make decisions based on them," but autonomy also needs to involve flexibility alongside agency to be able to address new and unexpected findings. The rule-based governing system, programmed by the human operator, makes single binary decisions that trigger an automated process. As such, the system would not be able to adapt to changes in chemistry or objectives without the operator redefining the rules. The application examples serve well to illustrate the automated robotics systems but are based on well-known chemistry.

The manuscript presents terrific automated system technology but does not include new scientific advances expected of a Nature contribution. Therefore, it seems a better fit for a more topic-specific journal, such as Nature Synthesis.

Comments:

1. It's unclear how the experiments conducted autonomously were selected. Were the decisions part of the autonomous workflow, or were reactions for each experiment preassigned?
2. Could more detail be provided about how the heuristic decision-maker works? From the main text, it appears to be a coded set of rules, which, of course, are interpretable but lack the flexibility to address new chemical applications. How was the inspector of spectra trained? For the overall software system, how many changes will be needed to do the next investigations?

3. It's stated that besides some restocking there was no human involvement during the experimentation. Can you comment on the required human involvement in setting up the experiments (setting the heuristic rules, planning experiments, setting goals, etc.) and analyzing final results (for example how was it determined that the one Sonogashira product cyclized, was it flagged by the NMR data? Is there a confidence score that the system reports to triage samples for human analysis? etc.). It would help the manuscript to make very clear everywhere a human is necessary (to claim autonomy).

4. The conclusion states: "This heuristic implementation also captures expert human knowledge ... which provides a focus for autonomous experimental searches in multi-dimensional chemical spaces that would otherwise be too large and too complex to navigate." Were any of the search spaces presented really too complex to do a full combinatorial experiment design? If you had a large search space, it would presumably be advantageous to be guided by desired properties.

5. The reader would be able to learn from the study if the authors had discussed insights gained in overcoming challenges in realizing the system and the demonstration examples rather than summarizing successful applications.

6. In many situations the heuristic fell back to using LC-MS and flagging NMRs for a human's interpretation. Ideally, every experiment would make use of both characterization modes, as that is practically the title of the paper, but it's often difficult to obtain interpretable hydrogen NMR spectra with the resolution limitations of low-field benchtop NMRs. It would have been helpful to have had a thoughtful evaluation of the utility of benchtop NMR analysis. Most pharmaceutical diversification efforts involve far more complex molecules than exemplified in the manuscript. Assuming that a high-field NMR unit was available in a neighboring lab, the robot could presumably transport the samples to the NMR input carousel. The paper could also highlight the need for tools to automatically interpret NMR spectra. Without trying to assign ^1H NMR peaks to specific protons, isn't finding the anticipated mass in the UPLC trace just as indicative of a reaction occurring?

7. Were two robots really needed? They seemed to be idle most of the time. Did having two robots lead to any acceleration in the rate of discoveries being made? Did having two robots require any additional considerations around scheduling their movements? Do the robots transferring samples between synthesis and analysis units provide advantages beyond the usual automation benefits, such as cost, 24/7 operation, and safety?

8. What was the rationale for selecting the chemistries? The pharmaceutical diversification reactions seem a little contrived. Only the Sonogashira coupling seems like its outcome might be up for question. Was any purification required between the different steps of the diversification campaign? If so, was it done automatically?
?

9. The scale-up is modest and simple, using slightly larger vials and multiple vials. In many practical situations, scale-up would be done in large flasks or even in flow that would require re-optimization of conditions. The main text has little detail on the procedure; it would be helpful to have some detail and discussion. Scale-up would also be an opportunity for the system to demonstrate agency

and flexibility.

10. Can the system accelerate the discovery of host-guest assemblies because it is better at picking compatible hosts and guests, or is it a matter of processing more samples? For the supramolecular chemistry, were the meaningful complex mass/charge ratios computed manually or automatically by the system? Were any new hosts synthesized? Were any new host-guest interactions found?

11. Which photocatalysts worked for the decarboxylative coupling?

12. The manuscript nicely references prior work, but the two recent reviews cited in the introduction (references 1 and 2) miss the latest reports on autonomous chemical synthesis in a rapidly evolving field (e.g., DOI: 10.1126/sciadv.adj0461, DOI: 10.1126/science.adi1407, DOI: 10.1039/D3EE02027D, DOI: 10.1126/sciadv.abo2626)

Minor comments/typos

1. If XRD is going to be explained in the methods it should be mentioned in the main text, otherwise move to SM.

2. Line: typo -> correction

118: can shared -> can be shared

224: catalyst -> catalysis

261: as for the -> as with the

620: requirements are failed -> requirements failed

646: Replicant -> Replicate (but Blade Runner is a great movie!)

Referee #2 (Remarks on code availability):

All the code is easy to find publicly available on Github in the Cooper Groups repository. In the manuscript they list the code as belonging to several Zenodo repositories, not all of the repos are publicly facing yet (like the raw data files), but that will presumably change at publication. The code looks fine, but we did not download and use it.

Referee #3 (Remarks to the Author):

This paper proposes a new automation concept for chemical synthesis. Rather than relying on a single measurement, it integrates both LC-MS and NMR to obtain reliable decision-making. They work in complementary ways to determine the success or failure of chemical reactions, and the approach has been exemplified with three application cases: diversification chemistry, supramolecular host-guest chemistry, and photochemical synthesis. This study deals with organic synthesis, and the scope of experiment has expanded compared to previous study (Ref. 14; Nature 583, 237–241, 2020), whose capability is formulation and characterization.

However, its technological progress and novelty are thought to be insufficient to be published in Nature as it is, even after revision, for several reasons. In terms of decision-making on synthesis

results, users should manually change heuristic rules for analysis whenever the chemical reaction or purpose of synthesis changes. Therefore, it can be said that there is no generalization effort for autonomous operation. And, when it comes to overall experimental workflow, this system is more of an “automation” than an “autonomous” because the workflow is open-loop type. Although the manuscript emphasizes the value of this research of “autonomous lab”, users have to intervene in various aspects, such as setting the heuristic decision-rules according to tasks and planning the next-round experiments. The title and contents should be revised to clarify the value and limitations of the study. Overall, it is difficult to say that there is a clear scientific progress other than employing a heuristic orthogonal analysis method.

I have several recommendations for the manuscript.

1. The title needs to be modified to describe well the contents and feature of the paper.
2. The words "automation" and "autonomization" are mixed in the manuscript. An accurate definition of the meaning of the two words is required, and the manuscript must be modified according to this definition.
3. The authors qualitatively describe the efficacy of the combined use of LC-MS and NMR. However, for a clear understanding, the accuracy of the pass/fail decision needs to be quantitatively summarized; true positive, true negative, false positive, false negative for each case of LC-MS, NMR, LC-MS and NMR.
4. Regarding the sentences of p10 “Diversification reactions that were deemed by the decision maker to be successful were purified by automated flash chromatography for isolation and for full characterization of the product, as is commonplace in medicinal chemistry discovery programmes”, there is detailed information about workup process including isolation of solid particles, hardware configuration, purification protocol, and isolation yield after the processes.
5. How is the reliability of the system, such as experimental accuracy or reproducibility?
6. The performance of 80 MHz-NMR may be limited. Were there any issues related to this to determine the synthesis result? If so, it will be necessary to discuss it in the manuscript.
7. In the SI, the reaction time is recorded for each product. How can the optimal reaction-time be determined?
8. In the SI (p5, p34, p45, p56), some chemicals appear to have been pre-weighted prior to the experiments. Why they were not dispensed automatically within the system during the process?
9. In the conclusion, some expressions of “application-agnostic” or “fully autonomous” seem to be exaggerations.

Author Rebuttals to Initial Comments:

We thank the three Referees for their detailed comments on the manuscript. While we disagree with some comments, the feedback is helpful. We feel that we failed to convey some aspects clearly enough, especially regarding the level of autonomy in the workflow.

Below is our response to the Referees' comments (responses in **blue**). To avoid repetitions, we start with clustered responses concerning what we considered to be three main criticisms made by more than one referee, that is: (i) use of dual robots; (ii) the level of autonomy, and; (iii) the choice of chemistry. After that, we provide a more detailed point-by-point response to the specific technical comments from the individual referees. We have also highlighted the associated **changes** in the manuscript that is attached here.

Response to the Core Objections from Referees 1–3

In our reading, the three main criticisms focused on (i) use of dual robots; (ii) the level of autonomy in the workflow, and; (iii) the choice of chemistry. The second point, autonomy, was questioned by all three referees, so we devote more space to that.

Criticism 1: Use of Dual Robots

This was commented on by **Referee #1** and **Referee #2**.

Referee #1: "(1) The reliance on two mobile robots rather than one is a disadvantage."

Referee #2: "Were two robots really needed? They seemed to be idle most of the time."

We feel that the demonstration of scalable autonomous platforms that might be deployed beyond individual academic labs is somewhat missing in this field. In this regard, we believe that the use of commoditized, mass-produced robots and standard lab equipment / consumables is attractive.

Our choice of two robots rather than one was made to demonstrate the potential scalability of our approach. However, as the Referees highlight, this does introduce equipment redundancy for the robots. We also pointed this out in the original submission ourselves. We have now made the relevant section clearer, more explicit, and briefer, as follows:

"We chose to use two task-specific mobile robotic agents to demonstrate the scalability of our approach into large industrial labs, which would comprise multiple synthesis platforms and, potentially, a much wider range of characterisation techniques. However, in the examples given here, this leads to significant equipment redundancy."

Future scalability aside, for the workflows that we present here, the Referees' criticisms are valid—the robots are idle for much of the time, and using more than one robot increases the cost. To address this, we have now shown that the workflow can be run using a single robot, as shown in a new Supplementary Video 5. To explain this, we made the following addition to the paragraph quoted above:

“We therefore also demonstrated that the mobile robot tasks can also be performed using a single mobile robot (Supplementary Video 5). We did this by designing a multipurpose robot gripper that could manipulate both the MS and NMR racks, along with the vials used for photocatalysis, each of which have quite different formats (Extended Data Fig. 2).”

These additions require the addition of a new co-author, Lyubomir Kotoplanov, who developed the new single-robot implementation, reflected in *Author Contributions* as follows:

“Z.Z and L.K. assisted with the programming of NMR-Agent and UPLC-Agent. R.C. designed, built, and installed custom hardware for the workflow. L.K. integrated the use of a single mobile robotic agent.”

We have also added the following section to the Supplementary Information:

“Initially, two mobile robots were used in the autonomous workflow, each capable of interacting with only one analytical instrument (NMR and LC-MS). The main difference between the mobile robots was represented by the gripper attached to the end of the robot arm. Each gripper was only compatible with its respective sample rack (NMR or LC-MS) and there was no cross-compatibility; that is, the NMR robot was only able to pick up NMR racks, and vice-versa.

To integrate the whole workflow under the operational capabilities of a single mobile robot, new LC-MS racks that were compatible with the “NMR gripper” were designed and produced via 3D printing. The gripper of the LC-MS robot was then replaced by another 3D-printed “NMR gripper”, such that the LC-MS robot could transport both the NMR racks and the newly designed LC-MS racks. The .STL files needed to 3D print the grippers and the custom NMR and LCMS racks are available as Supplementary Files.”

Criticism 2: Level of Autonomy / “Sophistication” of the Workflow

All three referees commented on this in one way or another, and we took this to be the strongest objection overall.

Referee #1: “The processes used to choose experiments seemed less sophisticated than prior work. It appears that the screening process simply comprised testing all available reactions.”

“In most papers describing the use of autonomous experimentation, the goal is to make better and smarter decisions than human researchers so that the system offers something beyond simply doing experiments faster.”

“The process of selecting which reactions to scale up had merit, but intelligence or decision making should be employed in selecting the experiments in the first place. Otherwise, it is not even clear this is a closed-loop process.”

Referee #2: “The advances seem incremental in view of the authors’ previous contributions and out of step with recent reports of autonomous synthesis based on integrated generative and property machine learning models. The article title includes the word, “autonomous”, but the manuscript does not describe to what extent autonomy is achieved.”

Referee #3: “In terms of decision-making on synthesis results, users should manually change heuristic rules for analysis whenever the chemical reaction or purpose of synthesis changes. Therefore, it can be said that there is no generalization effort for autonomous operation. And, when it comes to overall experimental workflow, this system is more of an “automation” than an “autonomous” because the workflow is open-loop type.”

Some of these points we feel are misconceptions, possibly because we did not explain the level of autonomy clearly enough without reference to the SI. Other points relate to varying definitions of “autonomous” versus “automated”, as discussed below.

First, we did not describe the system as “closed-loop”—the workflow is not an optimization process, unlike our 2020 catalysis paper—but it does involve autonomous, algorithmic decisions. We do not manually change the heuristic rules per reaction. Rather, we present three general chemistry problem types. These are specific to the type of *problem*, but not the details of the chemistry: for example, the supramolecular workflow might equally be applied to discovery of molecular knots or enzyme-like catalytic cages or macrocycles.

We note here that the term “closed-loop” is not synonymous with “autonomous”. There are already existing definitions here in the literature. For example, Jensen, Cronin, and Aspuru-Guzik (*Digital Discovery*, **2023**, *2*, 1259; *Nat. Commun.*, **2024**, *15*, 1240; *Digital Discovery*, **2024**, *3*, 842) have all made similar distinctions between “automation” as “the act of making a process occur without human intervention and autonomy as “a paradigm where feedback and adaptive decision-making afford the system agency over the manner of its actions.” Under these definitions, the system that we present is autonomous; it adapts according to feedback and makes autonomous decisions. Human involvement is limited to the input needed for setting up the platform and defining the objectives.

We have made several changes throughout the manuscript to clarify the level of autonomy of the platform and to make explicit which steps were performed by the humans; for example, in the schematic timelines now given in Figures 2b, 3b and 4b.

We have also summarised the human involvement and the division of tasks between robots and humans in this workflow in a new figure, Figure S 193:

Figure S 193. Division of tasks between humans and robots/algorithms in our workflow.

To make these points clearer in the main text, we have provided a schematic timeline for each of the three workflows in the updated parts **b** of Figures 2 and 3 and in part **d** of Figure 4, below:

Figure 2. Autonomous divergent syntheses. **a**, Overview of the synthetic route for used for autonomous parallel synthesis. The coloured chemical structures were successfully detected, while the greyed-out structures were not successful according to the rules of the decision maker. **b**, Heuristic decision-maker logic for screening, scale-up replication, and structural diversification steps, respectively. The timeline (bottom) shows tasks performed by humans in black boxes and tasks performed by the autonomous platform in green boxes. See Supplementary Information, Section 2, for details of reaction conditions.

Figure 3. Autonomous discovery of supramolecular host-guest systems. **a**, Combinations of amines, carbonyl-bearing pyridines, metal ions, and guests used for autonomous supramolecular host-guest cage syntheses. **b**, Heuristic decision-maker logic used for screening, scale-out replication, and host-guest binding experiments. In this case, the number of ^1H NMR peaks relative to the starting materials was used as a threshold criterion. The timeline (bottom) shows tasks performed by humans in black boxes and tasks performed by the autonomous platform in green boxes. See Supplementary Information, Section 3, for details of reaction conditions.

Figure 4. Addition of a photoreaction station to the modular, distributed workflow. **a**, Workflow design for integration an offline photoreactor station into the synthesis workflow. **b**, Autonomous screening of photocatalysts for decarboxylative conjugate addition. **c**, Photographs of screening samples being transferred into the SynLED photoreactor for irradiation. **d**, Decision maker logic for autonomous LC-MS data analysis. The timeline (bottom) shows tasks performed by humans in black boxes and tasks performed by the autonomous platform in green boxes. EY: Eosin Y; gCN: graphitic carbon nitride; 4Cz: 4CzIPN; Ir-1: $[\text{Ir}(\text{dtbbpy})(\text{ppy})_2]\text{PF}_6$, Ir-2: $(\text{Ir}[\text{dF}(\text{CF}_3)\text{ppy}]_2(\text{dtbpy}))\text{PF}_6$; TPT: 2,4,6-Triphenylpyrylium tetrafluoroborate. See Supplementary Information, Section 4, for details of reaction conditions.

Referee 2 suggests that our work is “out of step with recent reports of autonomous synthesis based on integrated generative and property machine learning models”. Looking again at the state-of-the-art in this field, we would disagree with this broad assessment, while accepting that there are emerging examples in specific domains of chemistry. As an example, we would

refer to the detailed discussion in Koscher *et al.*, “Autonomous, multiproperty-driven molecular discovery: From predictions to measurements and back” (*Science*, **2023**, 382, eadi1407; new ref. 17 in MS), and specifically section SM8/9/10 in the Supporting Information for that paper, titled “Human Involvement and Autonomy”. In our opinion, this study is one of the strongest recent examples in autonomous synthesis, albeit using a different automation approach. The paper is candid about the level of autonomy, for example (text from Koscher *et al.* in red):

Pg. 28: “Front End Predictions”

- Prediction Models (setup)
 - Selection of the properties of interest and discovery goal
 - Development of initial property models
 - Define and automate the model retraining strategy”

This is what we do in all three of our three workflows (e.g., black box in new Figure 3, “Humans Define Success Criteria”).

Pg. 29: Refill HPLC solvents

- The control network alerts the user when it predicts it will not have enough solvent to complete an operation. The user only needs to fill the containers and dismiss the alert for operations to resume.

This is what we define as ‘restocking’ in the new panels in Figs. 2–4.

Pg. 33: “The column was manually swapped between modes because the high-pressure switching valve that would enable automated column switching was backordered.”

We have no such interventions beyond restocking; the system is otherwise autonomous.

Pg. 30: In our workflow, a **human provides the scaffolds for the platform to explore**; however, molecules can be generated in a plethora of ways. A human was used to vet scaffolds for synthesizability and basal property profiles to save time by not having the platform try to make something fundamentally un-synthesizable or known to underperform. This does bias the platform, but does emulate the typical hit-to-lead workflow employed in the industry.

(Our bolding here.) This is precisely what we do in our study. Likewise, a more recent work by multiple authors (Strieth-Kalthoff *et al.*, *Science*, **2024**, 384, adk9227) elaborates around a specific, human-selected scaffold that relates to a molecule that was the best-performing organic laser in the literature hitherto. While Koscher *et al.* also incorporates some retrosynthetic tools in their workflow, which we did not, we would note that there is no comparable retrosynthesis approach for the open-ended supramolecular assembly workflow that we present here (Figure 3). This area of chemistry does not lend itself to retrosynthetic tools; they just do not work for supramolecular reactions of this kind. (See also comments in below regarding whether automated retrosynthesis is “smarter”).

Taking these points together, we feel that the autonomy of our platform matches recent leading examples in this field, and moreover we exemplify the approach over a broader range of chemistry problems. Supramolecular chemistry, such as in Figure 3, is hardly explored at all by autonomous approaches, and we would contend that the “*integrated generative and property*

machine learning models” invoked by Referee 2 do not exist in that context. Even for scaffold-based organic chemistry, as in our first example, we contend the distinction between our examples (Fig. 2) and the current state-of-the-art is narrower than implied. (See also response, below, to Criticism 3 re. relative novelty of the chemistry.)

Finally, Koscher *et al.* define an “Autonomy Score” in their paper, as below (pg. 32, ESI):

While the authors do not seem to score their own workflow, this is a helpful construct. We would score our platform as 2 in Adaptability, A (system chooses what methods to deploy from a fixed set based on measurement data) and 3 (the highest score) for Human Intervention, H. We cannot provide a score for Learning, L, since it is unclear how to frame this in the context of more open-ended molecular discovery workflows: this seems more relevant to closed-loop Bayesian optimization and other optimization methods, which is not our focus here.

Recent reviews by Aspuru-Guzik *et al.* and Abolhasani/Kumacheva addressing level of autonomy (*Digital Discovery*, **2024**, 3, 842 and *Nat. Commun.*, **2024**, 15, 1378) both refer to the work of Beal (*Mol. Syst. Biol.*, **2020**, 16: e10019) on autonomy classification for self-driving labs in synthetic biology, according to which our platform would score between 3 and 4 in term of autonomy (see figure below and text extract).

This paper categorizes our own previous work (*Nature*, **2020**, 583, 237) and Cronin's Chemputer platform (*Nature*, **2018**, 559, 377) as having a level 3 autonomy, as follows:

“For example, Level 3 autonomy has been demonstrated with organic synthesis via an integrated fluidic system and machine learning classifier (Granda et al, 2018) and in other chemical investigations via a mobile robotic system and Bayesian sample design (Burger et al, 2020). The “Adam” and “Eve” robotic science systems (King et al, 2009) arguably attain Level 4 autonomy, via systems biology knowledge representations that allow both experiment configuration from mechanistic hypotheses and hypothesis adjustment from results.”

Under these definitions, our new platform matches the level of autonomy of our earlier mobile robot example (Burger *et al.*, 2020) and in many aspects surpasses it: for instance, our new platform has much broader application scope, more refined decision-making capabilities, and enhanced ability to handle open-ended discovery workflows.

Reviewer 1 states that: “In most papers describing the use of autonomous experimentation, the goal is to make better and smarter decisions than human researchers so that the system offers something beyond simply doing experiments faster.”

We do agree that this is often the long-term goal of such studies, but we are less convinced that this is generally attained. Looking at the state-of-the-art in this area, some autonomous workflows, though by no means all, incorporate retrosynthetic tools. The work by Koscher *et al.*, above, is one such example. However, it is not clear that there are any examples, either in that paper or elsewhere, where retrosynthesis software leads to products that could not have been reached via human retrosynthesis, albeit perhaps more slowly. As such, workflows involving automated retrosynthesis might perhaps be faster, but it is more debatable that they are “smarter”. Similarly, in our examples, the automation of the analysis of NMR and UPLC-MS data saves large amounts of time in the workflow: the decision about the next experiments to perform is made by the algorithmic decision maker in milliseconds to seconds, irrespective of the time of day or night that the data is acquired. By contrast, human researchers would require significant analysis time to make such decisions, and further time to log and record the decision process – in that respect, we feel that our system makes “better” (more efficient) decisions but again, it is not necessarily “smarter”. Indeed, we would contend that human decision making by domain experts would be superior, albeit much slower. We are unconvinced that this is so different to other papers in this research area. A further relevant example would be the recent work by Szymanski, N. J. *et al.* (*Nature*, **2023**, 624, 86), where the decision making was highly autonomous, but not necessarily smarter or better than that of a domain expert.

While it is surely an aspiration of this field, we do not believe that the current state-of-the-art in this field is reflected by autonomous systems that can make smarter decisions than humans, and particularly not across a broad range of chemistries, as presented here. Indeed, there are recent high-profile examples to the contrary. We would argue that to achieve something like human parity for relatively complex decisions, which we believe we have demonstrated here, is a significant milestone in autonomy.

Criticism 3: The Chemistry Examples

Here the opinion was more mixed (our bolding).

Referee #1: “Now that this automated lab has access to these tools along with a more sophisticated chemspeed platform for synthesis, **it is capable of much more sophisticated and diverse chemistry.**”

Referee #2: “This dual characterization system was used to analyze the outcome of a series of drug-like molecule diversification reactions, host-guest interactions, guest assembly, and photocatalyst screening reactions. **The strengths of the study are the demonstration of diverse applications ...**”

However, the Referee also asks: “What was the rationale for selecting the chemistries? **The pharmaceutical diversification reactions seem a little contrived. Only the Sonagashira coupling seems like its outcome might be up for question.**”

Referee #3 does not really comment on the chemistry.

In fact, we would take the reverse view to **Referee #2** regarding the Sonagashira example. Metal coupling chemistry (e.g., Sonagashira & Suzuki coupling) has been a mainstay of the field of autonomous synthesis precisely because it is, by and large, not “up for question”. This chemistry is central to the two *Science* papers discussed above and the focus of work on Bayesian optimization by Doyle *et al.* (*Nature*, **2021**, 590, 89). In general, metal coupling chemistry is highly predictable and generalisable, our observed side reaction notwithstanding (Extended Fig. 4). Of course, no chemistry is totally generalisable and, as we pointed out, metal couplings failed for some substrates in our study; e.g., lack of formation of compounds **11** and **23** in Figure 2. Nevertheless, metal-coupled carbon-carbon bond forming reactions are some of the most generalisable synthetic reactions available, and it is therefore unsurprising that they have featured so strongly in the field of autonomous synthesis.

In strong contrast, our supramolecular examples have the potential to yield a wide array of products—this chemistry is *far* less predictable than metal coupling chemistry—which is where our two-factor NMR/LCMS method comes into its own. Moreover, some factors affecting host-guest binding (e.g., the influence of solvent) are almost entirely empirical, and thus highly suited for autonomous methods—that is, humans cannot make very smart *a priori* decisions, either, without measurement data.

As for the “rationale for selecting the chemistries”, the authors have, collectively, a longstanding interest in metal coupling chemistry (see e.g., *Angew. Chem. Int. Ed.*, **2007**, 46, 8574), supramolecular self-assembly (see e.g., *Nat. Mater.*, **2009**, 8, 973; *J. Am. Chem. Soc.*, **2018**, 141, 3605; *Angew. Chem. Int. Ed.*, **2015**, 54, 5636), and photocatalysis (see e.g., *Nat. Chem.*, **2024**, 10.1038/s41557-024-01546-5). As such, the choice of chemistry in all three examples is grounded in prior research from our team, although we would concede that the pharmaceutical example is based on a well-known and off-patent building block (Zidovudine was the first antiretroviral medication used to prevent and treat HIV/AIDS). We would maintain, however, that the basic diversification methodology is representative of current pharmaceutical practice, based on discussions that we have had with our industrial collaborators in that sector.

Point-by-Point Response to the Technical Questions from Referees 1–3

Referee #1 (Remarks to the Author):

The manuscript by Cooper et al describes the development and use of an autonomous lab for chemistry that involves the integration of two mobile robots. This is following the team's seminal paper in 2020 in which they showed that a single mobile robot is capable of designing and performing photocatalysis experiments. In some ways, this manuscript is an advance over the prior work. For instance, they report three independent workflows that leverage two or more analytical techniques, namely NMR and mass spectrometry. Now that this automated lab has access to these tools along with a more sophisticated chemspeed platform for synthesis, it is capable of much more sophisticated and diverse chemistry. The paper does demonstrate this through the three different types of chemistry.

We also feel this is the key advance; *Nature* 2020 was limited to benchtop photocatalysis. We thank the Referee for appreciating that our integration of multiple orthogonal analytical methods and more advanced synthetic tools gives access to sophisticated and previously uncharted spaces in automated organic chemical synthesis.

That said, in other important ways, this work represents a step backwards from the prior work. Specifically, while the workflows and results of the work appear technically sound, they are less impressive and impactful than the prior work in three ways:

(1) The reliance on two mobile robots rather than one is a disadvantage. While having different systems working together could be interesting if they interacted in novel ways, the two robots just represent parallel shuttles that allow samples to be delivered to two places. Further, it seems that one robot could have done both jobs if the team had designed a sample holder that worked with both analytical instruments. The idea that a new robot must be incorporated to accommodate each new hardware module is not scalable and contrary to the notion that the advances in this work should be in principle applicable to other labs.

As we explained in the manuscript, the use of two robots was a deliberate choice to demonstrate scalability to larger labs. However, it is not the case that a new robot is required for each new module; as demonstrated in this revision, we can perform the workflow operations with a single robot. This has now been addressed, above, in our response to Criticism 1.

(2) The processes used to choose experiments seemed less sophisticated than prior work. It appears that the screening process simply comprised testing all available reactions. This would quickly become impractical for parameter spaces of reasonable size. The process of selecting which reactions to scale up had merit, but intelligence or decision making should be employed in selecting the experiments in the first place. Otherwise, it is not even clear this is a closed-loop process.

As recognised by the referee, above, the experiments demonstrated in this work are more synthetically sophisticated than our prior work, requiring a wider range of measurements and laboratory techniques. The examples that we present here are open-ended chemical discovery reactions with a huge range of potential outcomes, particularly in the supramolecular workflow, which is quite distinct from tracking a uniform, well-defined GC peak in our previous work

(*Nature*, 2020). Since there is no fixed physical observable to optimise towards, this type of research is not amenable to optimisation algorithms, such as Bayesian Optimisation.

By pure necessity based on the potential and financial experimental cost, the search space explored here was smaller than our previous work. Similarly, other state-of-the-art optimization examples in synthetic chemistry (e.g., *Nature*, **2024**, 626, 1025) using high-throughput experimentation in well-plates have much smaller search spaces. Our system here targets semi-preparative scale synthesis, and for much larger spaces, the chemicals alone would be too expensive, at least for an academic programme.

The screening processes did not comprise testing all available reactions: this would be impossible given that there are, for instance, 2.1 million compounds reported in SciFinder that contain **24** as a substructure. Rather, the choice of search space was circumscribed by the human domain experts, as in other related studies (e.g., Koscher *et al.*, *Science*, **2022**, above). The goal of the present work was to provide chemists with a general synthetic platform that can be used to explore new chemistry with very limited human intervention (see response to Criticism 2, above). The selected chemistry examples stem from our domain expertise (see response to Criticism 3 above). Without such domain expertise in the planning stage, it is unclear how the initially selected reactions could not span a sufficiently rich and interesting search space, and we note that more open-ended ‘search everything’ strategies have met with mixed success (e.g., Szymanski, N. J. *et al.* *Nature*, **2023**, 624, 86).

As discussed above, our system is not a “closed-loop” process in the sense that it is not an optimisation problem, but we would maintain that its level of autonomy matches or exceeds other platforms in this field, as discussed in detail above (response to Criticism 2, above).

(3) In most papers describing the use of autonomous experimentation, the goal is to make better and smarter decisions than human researchers so that the system offers something beyond simply doing experiments faster. In this work, it appeared that the goal was to match the decision-making of humans. While it remains a virtue that they hit this target in terms of deciding which experiments to scale up, prior work has shown that autonomous systems can outperform human researchers, which makes the target set in this work a low bar.

We addressed this point in the response to Criticism 2, above. To build on that: we do agree with the referee that autonomous systems can outperform human researchers in certain chemistry problems; in particular, the case of optimising of highly specific, pre-defined research questions, especially in large multivariate search spaces (e.g., optimisation of reaction conditions to produce a specific chemical). However, such systems are still not able to match the decision-making of humans in more open-ended discovery process. Indeed, it has been suggested recently that sophisticated unsupervised AI-based algorithms for chemical discovery not only lack the explanatory power but might also lead to false attribution of novelty when uncoupled from domain expertise of human researchers (see e.g., *PRX Energy*, **2024**, 3, 011002). The close community scrutiny in that case stems from overreliance on unsupervised analysis of single-technique measurements; this is a key area that we address in our current work.

Another promise of autonomous synthesis is to increase the level of reproducibility and reliability, as well as being faster or (in some senses) 'smarter'. We show in our supramolecular workflow (Extended Figure 6), that successful hits are scaled out for replication six times, and the measurement data is checked for parity against the screening sample. Human researchers almost never perform this many replicate experiments; as such, we feel that the platform is outperforming human researchers in that sense.

These structural weaknesses aside, the team had a number of engineering achievements including the use of analytical tools without modification and fully automated workflows with minimal human involvement. Unfortunately, these do not seem sufficient to justify publication of this work in a high profile journal rather than a targeted engineering journal.

We would argue that we present a new scalable paradigm for autonomous chemical synthesis, demonstrated across a range of chemistry domains, and that this is much more than a set of engineering achievements. We hope that the revised manuscript now makes that point more clearly.

Referee #2 (Remarks to the Author):

Summary

This manuscript describes using dual modes of chemical characterization to enable a heuristic reaction analyzer to come to a better-informed verdict on the outcome of a variety of reactions. Each mode of chemical characterization is handled by a dedicated mobile robot that can shuttle reaction samples from a ChemSpeed ISynth to the benchtop NMR and UPLC-MS, respectively. This dual characterization system was used to analyze the outcome of a series of drug-like molecule diversification reactions, host-guest interactions, guest assembly, and photocatalyst screening reactions. The strengths of the study are the demonstration of diverse applications and automation of outcome decisions based on expert rules, all using characterization equipment that is being shared concurrently with human users.

We thank the referee for praising the generality of our approach to a wide range of chemical applications, as well the added benefit of modular automation where costly laboratory equipment is not hard-wired and monopolised by a single workflow.

The advances seem incremental in view of the authors' previous contributions and out of step with recent reports of autonomous synthesis based on integrated generative and property machine learning models.

Again, we would challenge this; see our comments above in response to Criticism 2. This study opens the use of modular mobile robotics to a new domain (organic synthesis). We are not aware of an uncontested example of 'integrated generative and property machine learning models' in this domain of autonomous synthesis across a range of synthesis problem types, as tackled here.

The article title includes the word, "autonomous", but the manuscript does not describe to what extent autonomy is achieved.

We believe we have addressed this in our detailed response to Criticism 2. We accept that we did not go into sufficient detail here in our first submission. We have now revised the manuscript in several areas, both in text and in figures, to make this clearer.

The authors state in the introduction that autonomy “requires agents, algorithms, or artificial intelligence (AI) to record and interpret analytical data and to make decisions based on them,” but autonomy also needs to involve flexibility alongside agency to be able to address new and unexpected findings.

The rule-based governing system, programmed by the human operator, makes single binary decisions that trigger an automated process. As such, the system would not be able to adapt to changes in chemistry or objectives without the operator redefining the rules. The application examples serve well to illustrate the automated robotics systems but are based on well-known chemistry.

We believe this has been addressed above (see response Criticism 2). “Well-known chemistry” here constitutes three completely different areas of chemistry, each with their own sets of rules. As stated above, the supramolecular example is unique in this domain, and we extend this to host binding assays, which are (at best) poorly understood, and certainly not well-known.

We would further argue that so far, most if not all examples in the field of autonomous synthesis could be categorized as being based on “well-known chemistry”. We also suggest that it would not be possible to present a fully “general” chemistry solution that can adapt autonomously to changes in chemistry or objectives. Jensen *et al.* (*Science*, **2023**, 382, 1374) recently highlighted that despite advances in robotics and AI, any autonomous or automated platform for chemistry still requires human intervention and input for initial goal setting and configuration. We do not believe that there are any generalised solutions, even within a specific chemistry domain, let alone across all chemistry.

Our workflow can handle “new and unexpected” findings, at least up to a point, because the heuristics are set prior to executing the workflow and are not set on a per-sample or per-substrate basis. For the screening stage itself, the NMR spectra of the product reaction mixture is compared by the system to the starting materials (provided to the decision-maker prior to the workflow starting as reference)—and the decision maker observes differences— as a human researcher would. It then takes ‘successful’ samples that meet our success criteria, set on a per-workflow basis, forward for replication by sending instructions to the ISynth synthesizer without human intervention or rule redefinition. Our scale-up and scale-out replication is set up precisely for the scenario of unexpected results, where it checks analytical data versus the screening stage, to observe and pseudo-quantify deviations using dynamic time warping in the case of NMR. The decision-maker rejects failed samples that deviate too much and then instructs the ISynth synthesizer not to carry those samples forward to the next stage, again without human intervention. In the authors’ view, this represents “agency” and “flexibility”, albeit guided by pre-programmed human-domain-expert rules.

The manuscript presents terrific automated system technology but does not include new scientific advances expected of a Nature contribution. Therefore, it seems a better fit for a more topic-specific journal, such as Nature Synthesis.

We thank the Referee for describing our advances as terrific. We also note that **Referee #1** recommended a purely engineering venue for the paper while **Referee #2** recommended a purely chemical venue (*Nat. Synthesis*), suggesting perhaps that our manuscript does have cross-domain importance to a wider general-science audience.

Comments:

1. It's unclear how the experiments conducted autonomously were selected. Were the decisions part of the autonomous workflow, or were reactions for each experiment preassigned?

We have now clarified the rationale behind the selection of the experiments and explicitly mentioned that the decisions were made by the autonomous workflow:

“This synthesis-analysis-decision cycle mimics human experimentation and decision processes to autonomously decide on the subsequent workflow steps. We exemplify the approach through structural diversification chemistry and the autonomous identification of supramolecular host-guest assemblies. While the syntheses were autonomous, the choice of chemistry was not: the reactions and building blocks were selected by domain experts prior to the experiments. This nonetheless gave a large reaction space for the decision-maker to navigate—for example, in the case of the supramolecular reactions, there were many plausible stoichiometric assemblies. We also extended this autonomous approach beyond synthesis to assay function, for example by autonomously assessing the host-guest binding properties of successful supramolecular syntheses.”

2. Could more detail be provided about how the heuristic decision-maker works? From the main text, it appears to be a coded set of rules, which, of course, are interpretable but lack the flexibility to address new chemical applications. How was the inspector of spectra trained? For the overall software system, how many changes will be needed to do the next investigations?

The decision maker rules are outlined in Figures 2–4 and accompanied by walk-through examples in Extended Data Figures 3, 5, 6. Due to space constraints, we decided to limit the details about the decision maker to figures in the Supplementary Information. Indeed, the SI (pp. 5–125) summarised every reaction performed by the workflow alongside detailed expert chemist assessments.

To make this clearer without reference to the SI, the revised versions of Figures 2–4 now highlight all decision-making criteria and the settings that are selected prior to execution of the workflow – in our view, each workflow is therefore autonomous according to common definitions (see our general response to Criticism 2).

The accompanying Zenodo repository contains all the data required to reproduce the analysis (raw data acquired by the autonomous workflow) and each analysis method used by the decision maker exemplified step-by-step in the Jupyter notebook. We have ensured now that all data files and Jupyter notebooks are public-facing. We also deposited code examples on Zenodo (submitted with the manuscript) that illustrate just how few changes are needed

between investigations — essentially, the decision criteria can be deemed universal across a wide landscape of chemistry reaction types, at least for solution-based organic chemistry. Our decision-maker is built around a highly general core set of rules, requiring only parameter adjustments to apply it to different types of chemistry. This means that the entire decision-maker does not need to be rewritten for each case.

As detailed in the code files, the “spectra inspector” (a nice name!) uses standard TopSpin and SciPy peak picking algorithms. We do not deem it necessary to employ more sophisticated ML-based algorithms for identification of changes between XY spectra files. For comparison, recent work by Cronin and others (*ACS Cent. Sci.*, **2021**, 7, 12821) used data from 440 reactions to train a CNN model that was able to predict the correct reactivity class (based on the number of peaks that have appeared in the reaction mixture as compared to the starting materials) only 56% of the time. Our simple and explainable heuristic method is not only more accurate but can be easily extended to contain more sophisticated examples, for instance to identify a reaction as “incomplete” by presence of starting material peaks and a set of peaks corresponding to a new partially formed product. It is also considerably more efficient as it does not require the acquisition of an extensive training set of reactions (which is both time consuming and expensive) for each new type of chemistry explored.

We have now included time estimates in the new Figure S193 (see above) to show how simple it is to adapt the general decision-making methods from one workflow to another. We believe that this further strengthens the argument that well-designed heuristic rules are easily transferable between different areas of chemistry by domain experts.

3. It's stated that besides some restocking there was no human involvement during the experimentation. Can you comment on the required human involvement in setting up the experiments (setting the heuristic rules, planning experiments, setting goals, etc.) and analyzing final results (for example how was it determined that the one Sonogashira product cyclized, was it flagged by the NMR data? Is there a confidence score that the system reports to triage samples for human analysis? etc.). It would help the manuscript to make very clear everywhere a human is necessary (to claim autonomy).

This has now been fully addressed above and reflected in the manuscript (see general response to Criticism 2).

The cyclized product was indeed identified by human inspection of the NMR data offline, following purification for unambiguous structural confirmation. This is now reflected in the revised manuscript:

*“While target molecules **14** and **15** were successfully synthesized, **13** underwent an unexpected intramolecular cyclization reaction (Extended Data Fig. 4). This was identified by human inspection of the NMR data and confirmed through single crystal x-ray diffraction measurements.”*

4. The conclusion states: “This heuristic implementation also captures expert human knowledge ... which provides a focus for autonomous experimental searches in multi-dimensional chemical spaces that would otherwise be too large and too complex to navigate.” Were any of the search spaces presented really too complex to do a full combinatorial

experiment design? If you had a large search space, it would presumably be advantageous to be guided by desired properties.

All three areas of chemistry studied are much too large for brute force screening. For example, as noted earlier, there are 2.1 million structures reported in SciFinder that contain compound **24** as a substructure. This is too complex for a full combinatorial experiment design, and the use of human knowledge to frame the search space is essential. An alternative and perhaps valid strategy would have been to use some form of chemical diversity algorithm to limit the search space, rather than human domain expert knowledge, which could have the virtue of reducing human bias. We contend that there is no general answer to this question—that is, open-ended, computer-driven design versus human planning—but we felt that the human-focused choice was strongest for the problems presented here, and it is the dominant paradigm in this field so far.

We also refer the Referee to our response to **Referee #1**, where we discuss the difficulty of applying properties-based optimization methods to guide open-ended discovery campaigns with multiple possible products, for which there is no fixed physical observable to optimise towards. We are not aware of any other system that can make even a qualitative decision on the success of supramolecular assembly without any human domain expertise.

5. The reader would be able to learn from the study if the authors had discussed insights gained in overcoming challenges in realizing the system and the demonstration examples rather than summarizing successful applications.

This could be interpreted as referring to the automation aspects, the chemistry, or both; we read it as the latter.

(i) Automation challenges: To our knowledge, we, along with Jensen's group (*Science*, **2023**, 387, 6677) are the only research groups in autonomous chemistry that have attempted any analysis of failure modes in our robotic set ups (see *Nature* 2020, ESI Fig. S37). In this new study, there were no robotics errors in these experiments because of extensive technical developments since 2020, but we again openly discussed other errors that happened during the workflow (such as two WiFi disconnections) in the SI (section 5.1).

(ii) Chemistry challenges: We report and discuss all unsuccessful applications performed by the system and comment on them extensively in the Supplementary Information. Just to give just two examples: supramolecular screening reaction 9 (SI section 3.2.9) formed a supramolecular structure, but its nature could not be determined, even by a trained chemist, and this is discussed. We also report the failed synthesis of “simple” compound **11**, where traces of desired compound were seen but the conditions would need to be heavily optimised to achieve full conversion.

In addition to this comprehensive commentary in the Supplementary Information, we also included several qualifying statements in the manuscript that we felt gave a balanced view of the advantages and limitations of our method, and the associated challenges, for example (see **red** text in particular):

“We chose to use two task-specific mobile robotic agents to demonstrate the scalability of our approach into large industrial labs, which would comprise multiple synthesis platforms and, potentially, a much wider range of characterisation techniques. However, in the examples given here, this leads to significant equipment redundancy.”

“While target molecules **14** and **15** were successfully synthesized, **13** underwent an unexpected intramolecular cyclization reaction (Extended Data Fig. 4). This was identified by human inspection of the NMR data and confirmed through single crystal x-ray diffraction measurements. This species has the same molecular weight as the uncyclized cross-coupling product and was not distinguishable by UPLC chromatograms or MS alone. This highlights both the need for orthogonal characterization methods, and the limitations of autonomous decision making for such unexpected edge cases.”

“One technical limitation was that the high dispersion related to the low field strength of the benchtop NMR instrument (80 MHz) can lead to an artificial increase in the apparent number of peaks.”

“Of course, such pre-programmed rules also introduce confirmation biases, and might miss important new reactions, but the algorithmic workflow is fully traceable⁵ and the data for all reactions, including ‘unsuccessful’ ones, are saved for future inspection.”

6. In many situations the heuristic fell back to using LC-MS and flagging NMRs for a human’s interpretation. Ideally, every experiment would make use of both characterization modes, as that is practically the title of the paper, but it’s often difficult to obtain interpretable hydrogen NMR spectra with the resolution limitations of low-field benchtop NMRs. It would have been helpful to have had a thoughtful evaluation of the utility of benchtop NMR analysis. Most pharmaceutical diversification efforts involve far more complex molecules than exemplified in the manuscript. Assuming that a high-field NMR unit was available in a neighboring lab, the robot could presumably transport the samples to the NMR input carousel. The paper could also highlight the need for tools to automatically interpret NMR spectra. Without trying to assign ¹H NMR peaks to specific protons, isn’t finding the anticipated mass in the UPLC trace just as indicative of a reaction occurring?

This is a valid point, and we agree that the direction of travel in pharmaceutical chemistry is toward larger, more complex molecules, although this is only one of the three examples of chemistry that we discuss in the paper. We also commented on the constraints imposed by using benchtop NMR ourselves. Indeed, we discussed at length the technical limitations of low-field NMR spectrometer in the original manuscript. However, we also presented the benefits of using a low-field benchtop NMR spectrometer, such as:

- Use of purely non-deuterated solvents and the associated costs around £50 on Merck for 5 grams of acetonitrile-d₃); even on high-field systems solvent suppression would not improve the reliability of the decision maker.
- No need for sophisticated sample handling (spinners, etc.).
- No need for cryogenic cooling of the probe and magnet: liquid helium and liquid nitrogen refills are not needed on regular weekly basis in our setup).
- No need for shimming to the sample as this is done to a standard sample with accurate algorithms.

- No need to the presence of any deuterated species in the sample as locking is done to an external capillary, further lowering the cost and number of manipulations in high-throughput experimentation.
- Ease to retro-fit in existing laboratory

In fact, the vast majority of reported autonomous synthesis platforms so far use some form of benchtop NMR, often using a flow system (unlike our implementation), and those systems will have the same disadvantages and benefits as discussed above. One exception is the ongoing efforts of the SWISSCAT project in Switzerland (*CHIMIA* **77**, 154, 2023), which aims to incorporate a high-field NMR spectrometer into an automated laboratory setup. This impressive feat is being realised using relay-type sample handling that involves four specialised robots and extensive laboratory engineering. As such, while high-field NMR integration is possible, it is likely to be expensive and complex. By comparison, no such modifications were required in our work and the only modification was to design a new sample holder that could be manipulated by the robot. We have now added the following sentence to reflect this:

“While we used a benchtop NMR here, there are situations where the performance of high-field NMR may be necessary, for example to characterize large, more complex pharmaceutical molecules. However, there are many technical barriers to integrating high-field NMR into automation workflows, and while efforts are being made to accomplish this, it introduces significant customization, cost, and complexity, perhaps limiting it to more specialized facilities.”

The Referee’s point regarding complex pharmaceutical molecules is important, and some high molecular weight drug targets can be problematic even for higher-field NMR. We see two options here for the future. First would be to introduce automated high-field NMR, although as discussed above, that is likely to be highly complex and expensive. The second strategy would be to rely less on NMR and to introduce new, complementary methods into our workflow, such as more advanced hyphenated chromatographic techniques, which is already a trend in the pharmaceutical industry because of the growing complexity of drug targets, as the Referee rightly highlights (see e.g., *Nat. Rev. Chem.*, **2019**, 3, 4-14, and *Science*, **2024**, 383, 612, and *Anal. Chem.* 2019, 91, 1, 210–239 - for IM-MS in supramolecular chemistry, enantioselective separations, and chromatography in pharmaceuticals generally). Here, the modular and expandable nature of our approach would be a strong advantage.

We did explore the use of existing commercial tools for automated assignment of NMR spectra but found that they were either unsuitable (particularly for the supramolecular workflow) or performed poorly, particularly in reaction mixtures of unpurified compounds.

With respect to the question of whether UPLC-MS alone was sufficient, we gave multiple examples in the Supplementary Information (Section 3.2) and in Extended Figure 5 where MS measurements were ‘passed’ but the NMR analysis was a ‘fail’. Many ‘drug-like’ molecules are easily ionized and separate well on liquid chromatography, and in this case UPLC-MS alone might be sufficient. This does not work nearly as well for areas such as our supramolecular chemistry example, nor does UPLC-MS provide any information on host-guest binding.

7. Were two robots really needed? They seemed to be idle most of the time. Did having two robots lead to any acceleration in the rate of discoveries being made? Did having two robots require any additional considerations around scheduling their movements? Do the robots

transferring samples between synthesis and analysis units provide advantages beyond the usual automation benefits, such as cost, 24/7 operation, and safety?

The point regarding dual robots and equipment redundancy has been addressed above (see response to Criticism 1)—we can operate the workflow with a single robot, if needed. There would be a significant acceleration in using multiple robots for larger workflows, but not in the case presented here.

The point Referee's question about cost and 24/7 operation, which will define the return on investment, is important. We chose not to discuss cost and logistics in the first submission, but we feel that this might ultimately dictate the uptake of these technologies. In response, we have now added a fuller discussion in the concluding paragraphs, as follows:

“The best ratio of robots to other equipment will depend on the setting. Our lab contains just one ISynth platform, but by extension, multiple Chemspeed platforms, or other synthesis platforms, could be serviced by a team of mobile robots in an assembly line process. The approach should therefore be scalable into the largest industrial laboratories, if necessary connecting physically separated synthesis and analytical labs using mobile robots that can traverse buildings⁵⁹. In such a distributed scenario, the cost of the mobile robots might be a relatively minor consideration with respect to the instruments that are served, particularly since industrial mobile robots are a highly commoditized technology⁶⁰ that serves multiple sectors beyond chemistry, such as automotive manufacture and warehousing.”

To put this in specific context: for the workflow illustrated here, the combined cost of the synthesis platforms and the analytical modules was around £1.4M—the UPLC-MS alone cost £260K. By contrast, a single KUKA mobile robot cost around £130K at the time of purchase. These industrial robots are highly commoditized and used in multiple sectors; as such, they are unlikely to be the cost-limiting factor in synthetic workflows of this type. Already, one mobile robot represents <10% of the overall cost of the instrumentation used in our examples, and unlike bespoke chemistry hardware, all forecasts that we are aware of predict that the cost of industrial robots will fall further in the future, mainly because of cross-sector price scaling in this huge global market (see e.g., https://www.ey.com/en_tw/innovation/three-tailwinds-for-robotics-adoption-in-2024-and-beyond: “The average price of an industrial robot has halved over the past decade, to about US\$23,000 in 2022 from US\$47,000 in 2011, according to ARK Invest, which predicts that costs will fall a further 50% to 60% by 2025”).

8. What was the rationale for selecting the chemistries? The pharmaceutical diversification reactions seem a little contrived. Only the Sonagashira coupling seems like its outcome might be up for question. Was any purification required between the different steps of the diversification campaign? If so, was it done automatically?

We address this above in our combined response to Criticism 3.

No purification was done in the intermediate stages. We agree that this would be a valid future target for extending the methodology, and it is indeed a technical limitation of our currently set-up (and to our knowledge, almost all other autonomous platforms).

9. The scale-up is modest and simple, using slightly larger vials and multiple vials. In many practical situations, scale-up would be done in large flasks or even in flow that would require re-optimization of conditions. The main text has little detail on the procedure; it would be helpful to have some detail and discussion. Scale-up would also be an opportunity for the system to demonstrate agency and flexibility.

We feel that this is context dependent. In synthetic organic chemistry laboratories, scale-up of 5x reaction size is usually considered to be more than “modest and simple” because the reaction outcomes might change substantially upon such a scale-up. For example, see <http://www.chem.rochester.edu/notvoodoo/pages/reaction.php?page=planning>, which states that:

“If you have run the reaction before, you may choose to run a larger scale reaction. It is best to scale up by no more than 3–4 times the previous experiment, in case the reaction begins to lose efficiency.”

While even larger scale-up and process optimization would certainly be an interesting target in the future, we would consider it beyond the scope of this study.

We would argue that we have demonstrated flexibility and agency: the decision-maker checks analytical data from scale-up reactions for parity with the screening samples, and rejects scale-up experiments that are not consistent with the screening results *without human intervention*. As such, the decision maker would not take “failed” scale-ups forward in the diversification stage. We refer to Extended Figure 3 and the Supplementary Information Section 2.3 (for scale-up reaction conditions) and Section 2.4 (for consistency testing) for further details.

10. Can the system accelerate the discovery of host-guest assemblies because it is better at picking compatible hosts and guests, or is it a matter of processing more samples? For the supramolecular chemistry, were the meaningful complex mass/charge ratios computed manually or automatically by the system? Were any new hosts synthesized? Were any new host-guest interactions found?

The system did not pick compatible hosts and guests, but it fully automates the laborious trial-and-error process that is commonly employed in the discovery of such systems. To date, no clear rationale exists to predict with precision host-guest compatibility, and in most cases, empirical observation remains the main strategy. Even expensive, high-level computational approaches often fail because of the difficulty of accounting for solvent and/or host flexibility.

We chose a selection of building blocks with varying steric and electronic effects that could potentially form hosts, and using domain expert knowledge. Furthermore, some of the chosen guests were expected to exhibit fast and slow exchange on the NMR timescale, thus covering the two usual cases encountered in host-guest chemistry. We also made sure that both aliphatic and aromatic compounds were represented. This was done to test the limits of automated detection of successful binding interactions by our benchtop NMR setup. We have now explained this choice in the manuscript:

“Some of the example guests were expected to exchange slowly on the NMR timescale.”

As detailed in the associated code, the mass-to-charge ratios were automatically enumerated with different scripts: directly by the workflow for the organic synthesis outcome adducts or as part of experimental setup for the supramolecular discovery workflow. Those procedures are included in the associated repositories and they can be easily adapted to other problems.

We were cautious about claiming compounds discovered to be “new” or novel, since this is a hot topic in this field. Crystal structures of the exact host complexes reported in this work were not previously deposited in the Cambridge Structural Database and could be referred to tentatively as “novel”, although related structures have been published previously, so we chose not to stress that novelty. Likewise, most compounds reported in the organic diversification workflow had not been previously reported—they are new chemical entities—but we would not claim that they are particularly novel or unusual in structure.

No fundamentally new host-guest interactions were found in this study: most of those assemblies are dominated by solvophobic effects and dispersion interactions. However, the binding of cyclohexanol inside $[Zn_4(\mathbf{24}_3, \mathbf{28})_4]^{8+}$ has not been reported previously. Even though this technically constitutes a novel discovery, we again did not deem it important enough to claim in the manuscript since binding of this guest had been observed in a related Fe-based capsule that is already cited in our report (ref. [50]). That said, it is by no means always true that host-guest properties of metal-organic cages remain the same when the metal ions are changed, so the binding of cyclohexanol inside $[Zn_4(\mathbf{24}_3, \mathbf{28})_4]^{8+}$ is not obvious *a priori*.

11. Which photocatalysts worked for the decarboxylative coupling?

We have now amended the manuscript to specify which photocatalysts were successful, as per the existing information from Section 4.2.2 in the Supplementary Information.

“Three catalysts (4Cz, Ir-1, Ir-2) were found to yield the desired decarboxylative conjugate addition product. The other three photocatalysts (EY, gCN, and TPT) and a blank control produced only starting materials, with no trace of the product.”

12. The manuscript nicely references prior work, but the two recent reviews cited in the introduction (references 1 and 2) miss the latest reports on autonomous chemical synthesis in a rapidly evolving field (e.g., DOI: 10.1126/sciadv.adj0461, DOI: 10.1126/science.adi1407, DOI: 10.1039/D3EE02027D, DOI: 10.1126/sciadv.abo2626)

As requested, we have added the following references in the text (10.1126/sciadv.adj0461 was already cited as reference [13]) and renumbered the other references accordingly:

[17] *Science*, **2023**, 382, adi1407 (10.1126/science.adi1407).

[18] *Matter*, **2024**, 7, 2382 (10.1016/j.matt.2024.06.003).

[26] *Sci. Adv.*, **2022**, 8, abo2626 (10.1126/sciadv.abo2626)

[27] *Energy Environ. Sci.*, **2023**, 16, 3984 (10.1039/D3EE02027D)

Minor comments/typos

1. If XRD is going to be explained in the methods it should be mentioned in the main text, otherwise move to SM.

Single crystal XRD structures are included in the Extended Data Figures 4 and 5, which we understand to form part of the main text. Therefore, we felt that the refinement methods should be summarised and cited in the main text. If this is not in line with Editorial guidelines, we are happy to move this information to the Supplementary Information.

2. Line: typo -> correction

118: can shared -> can be shared

224: catalyst -> catalysis

261: as for the -> as with the

620: requirements are failed -> requirements failed

646: Replicant -> Replicate (but Blade Runner is a great movie!)

Thanks - those have now been corrected.

Referee #2 (Remarks on code availability):

All the code is easy to find publicly available on Github in the Cooper Groups repository. In the manuscript they list the code as belonging to several Zenodo repositories, not all of the repos are publicly facing yet (like the raw data files), but that will presumably change at publication. The code looks fine, but we did not download and use it.

We have now checked that all repositories linked in the manuscript are publicly facing. It seems that the raw data from our experiments have already been downloaded over 50 times, even though no link was published anywhere beyond this manuscript.

Referee #3 (Remarks to the Author):

This paper proposes a new automation concept for chemical synthesis. Rather than relying on a single measurement, it integrates both LC-MS and NMR to obtain reliable decision-making. They work in complementary ways to determine the success or failure of chemical reactions, and the approach has been exemplified with three application cases: diversification chemistry, supramolecular host-guest chemistry, and photochemical synthesis. This study deals with organic synthesis, and the scope of experiment has expanded compared to previous study (Ref. 14; Nature 583, 237–241, 2020), whose capability is formulation and characterization.

However, its technological progress and novelty are thought to be insufficient to be published in Nature as it is, even after revision, for several reasons. In terms of decision-making on synthesis results, users should manually change heuristic rules for analysis whenever the chemical reaction or purpose of synthesis changes.

Therefore, it can be said that there is no generalization effort for autonomous operation. And, when it comes to overall experimental workflow, this system is more of an “automation” than an “autonomous” because the workflow is open-loop type.

We believe this is incorrect and we refer to our combined response to Criticism 2 (above). The heuristic rules are only adjusted on “**per workflow**” basis, not per individual reaction or sample. Also, **only one set of general rules are used throughout each workflow**, which is set at the start of the workflow. We feel that the inclusion of more than one rule set is reasonable since each workflow tackles a completely different domain of synthetic chemistry. To make this any

broader—that is, to encompass all possible domains of chemistry in one rule set—would simply be unphysical.

We also note that similar criticisms can be levelled at optimization techniques, such as Bayesian optimization. It is common practice to adjust the priors to a specific problem type (*i.e.*, manually change the rules) and the objective function is frequently set by hand prior to the experiment. Jensen *et al.* (*Science*, **2023**, 382, 1374), recently highlighted that despite advances in robotics and artificial intelligence, all autonomous or automated platforms for chemistry still require human intervention and input for initial goal setting and configuration.

We are unaware of any *bona fide* examples of an autonomous synthesis platform or experimental decision-making algorithm that can investigate completely different domains of chemistry, or completely different objectives (*e.g.*, going from screening to scale-up to diversification, as demonstrated in this study) without any human input. To the best of our knowledge, there are no reported examples approaching the Referee's desired level of autonomy.

Although the manuscript emphasizes the value of this research of “autonomous lab”, users have to intervene in various aspects, such as setting the heuristic decision-rules according to tasks and planning the next-round experiments.

As we pointed out at multiple points in the manuscript, it is the algorithm and the data that plan the next-round experiments, not the researcher (see response to Criticism 2, for extensive discussion). Humans only define the success criteria for each step **before the experiment**, as in all other published examples of autonomous chemistry (*e.g.*, with Bayesian methods) but there is **no** intervention during the workflow execution beyond the initial set-up. We have now modified the figures in the manuscript to reflect this and added a specific section in the Supplementary Information discussing the “level of autonomy” of our platform.

The title and contents should be revised to clarify the value and limitations of the study. Overall, it is difficult to say that there is a clear scientific progress other than employing a heuristic orthogonal analysis method.

This view is not reflected by the other two referees, who did find examples of scientific progress in the work.

I have several recommendations for the manuscript.

1. The title needs to be modified to describe well the contents and feature of the paper.

We have dropped the word “Twin” from the title, since we now show that the workflow can be accomplished using a single mobile robot. As such, the title reads:

“Cooperative Mobile Robots for Autonomous Synthetic Chemistry”

We have argued in detail, above, as to why we believe this platform to be autonomous. The robots are mobile. “Cooperative” refers to the fact that the robots can share equipment with human researchers without monopolizing it, which we feel is a key distinction of our approach. While we do not contend that our platform could perform all synthetic chemistry—it could not do high-pressure reactions, for example—we feel that the inclusion of three different examples

is broad enough to use the term “Synthetic Chemistry”, and anything more specific (e.g., drug discover, supramolecular chemistry) would not reflect the contents of the paper. This could in principle be modified to “Synthetic Organic Chemistry”, since all the examples given are organic (or metal-organic), and we would follow editorial advice on such points.

2. The words "automation" and "autonomization" are mixed in the manuscript. An accurate definition of the meaning of the two words is required, and the manuscript must be modified according to this definition.

We have checked our manuscript and found no mention of ‘autonomization’. We are content with the definitions provided by others regarding the distinction between “automated” and “autonomous” (see papers cited above in response to Criticism 2) and we do not feel it would be helpful to redefine those terms ourselves.

3. The authors qualitatively describe the efficacy of the combined use of LC-MS and NMR. However, for a clear understanding, the accuracy of the pass/fail decision needs to be quantitatively summarized; true positive, true negative, false positive, false negative for each case of LC-MS, NMR, LC-MS and NMR.

The Supplementary Information provides a justification/overview for how the decision maker performed for every experiment in both the parallel synthesis (Section 2.2 - 2.8) and the supramolecular chemistry workflows (Section 3.2). We also provided a ‘chemist comment’ for each reaction in the supramolecular screening stage and compare the data to a manual experiment with high-field NMR data.

4. Regarding the sentences of p10 “Diversification reactions that were deemed by the decision maker to be successful were purified by automated flash chromatography for isolation and for full characterization of the product, as is commonplace in medicinal chemistry discovery programmes”, there is detailed information about workup process including isolation of solid particles, hardware configuration, purification protocol, and isolation yield after the processes.

The workup details can be found in the Supplementary Information for the relevant reactions (SI Section 2.5 and 2.7). The purification was done offline in the final stage. We performed offline purification of aliquots of the reaction mixture for unambiguous structural confirmation of the products on high-field NMR and high-resolution mass spectrometry. We have now specified the system used for flash purification in the SI (SI Section 2.5 and 2.7). LC area percentages for the reaction mixtures are available in the Supplementary Information along with each product.

5. How is the reliability of the system, such as experimental accuracy or reproducibility?

We refer to Extended Figure 6 in our supramolecular workflow where we provide comparison of experimental data between multiple reaction replications, as well as the Supplementary Information (Section 3.4).

6. The performance of 80 MHz-NMR may be limited. Were there any issues related to this to determine the synthesis result? If so, it will be necessary to discuss it in the manuscript.

See NMR response to **Referee #2**, point 6; we summarised this already in the manuscript. Similarly, assessment of all spectra is already included in the Supplementary Information.

7. In the SI, the reaction time is recorded for each product. How can the optimal reaction-time be determined?

The reaction times reported were the actual times that the reagents were allowed to react for. No optimal reaction-time was determined, nor is it common practice in exploratory synthetic batch chemistry research. Reaction time optimization could be a future application of our system in the future, but this is beyond the scope this study.

8. In the SI (p5, p34, p45, p56), some chemicals appear to have been pre-weighted prior to the experiments. Why they were not dispensed automatically within the system during the process?

While we could have dispensed these automatically, we were constrained by space on the deck of the synthesizer since the reaction vials cannot be individually heated on the ISynth platform, and heating concentrated stock solutions of isocyanates and amines would have been unsafe. Furthermore, solid dispensing solutions on the ISynth platform is possible but inefficient at smaller scales. For instance, entire dispensing cartridges would need to be pre-packed by human researcher in gram quantities to then dispense just 100 mg. Hence, the use of automated solid dispensing did not present any advantages in this case because human intervention was required to load the dispensing cartridges, as well as needing much more starting material. We note that the solids did not need to be accurately pre-weighed: so long as the weighted mass was recorded, the Chemspeed software adjusts solvent volumes to make up solutions at appropriate concentrations. This represents a significant time saving in terms of setting up the platform.

9. In the conclusion, some expressions of “application-agnostic” or “fully autonomous” seem to be exaggerations.

We would defend our use of the term “application-agnostic”. Our platform is applicable to a large portion of solution-phase synthetic chemistry that involves analysis by NMR, chromatography, and mass spectrometry.

We did caveat our statement about fully autonomous, as follows:

“The workflows are fully autonomous, apart from some limited manual human intervention to restock chemicals and consumables,”

With hindsight, though, we agree with the referee on this point – no system reported so far, ours included, is “fully autonomous” because human researchers are setting the objectives, as pointed out recently by Jensen *et al.* (*Science*, **2023**, 382, 1374). We have now revised this sentence as follows:

“The workflows are autonomous once the objectives have been set, apart from some limited manual human intervention to restock chemicals and consumables, ...”

Reviewer Reports on the First Revision:

Referees' comments:

Referee #1 (Remarks to the Author):

The authors have provided a thorough rebuttal to my comments and the comments of the other Referees, but unfortunately upon re-reading the manuscript, I am left with the perception that the present work is mainly a robotics and integration advance with a step backwards in terms of how automation is used to change how research is done. The simplicity of the decision making is such that a team of trained undergraduates could replicate this work in a similar amount of time by following a simple check list. The goal of automated labs is not to replace researchers, but to enable new types of exploration. For this to happen, the knowledge from experiments must be meaningfully incorporated into subsequent experiments more than just knowing whether that particular experiment is worth repeating at a larger scale. While the authors state that closed-loop operation is only currently possible in optimization, what is discovery but optimization of novelty in a large parameter space. I reject that automated curve analysis is a meaningful level of agency for a research robot of this scale. I agree with Referee #2 that a journal such as Nature Synthesis is a better fit for this work although Nature Chemical Engineering could be appropriate given the focus on robotics. Either way, this work does not comprise a transformational advance but rather an incremental one.

Referee #2 (Remarks to the Author):

The authors compellingly addressed and rebutted most of the comments raised in the initial review. The acknowledgement of human input for initializing each task and fine-tuning the analysis heuristics/thresholds for each task clarifies the claim of autonomous synthetic chemistry. The additional text explaining the demonstrative purpose of two cooperative robots when only a single robot is necessary for the chosen tasks addresses a substantial apparent shortcoming in the original manuscript. The argument that the chemistry explored is diverse and difficult to search with learning or optimization approaches is persuasive, especially regarding supramolecular assembly where there is not yet enough understanding to build predictive synthetic tools.

The main concern is that the study is a demonstration of an open-ended exploration that would have been successful no matter the outcome of the chemical studies. There is no benchmark to achieve or hypothesis to test. While the revised manuscript and rebuttal repeatedly state the project does not demonstrate solving an optimization problem, experimental design involves either hypothesis testing or some optimization challenge. Perhaps emphasis could be added on what new knowledge was gained in this study, be it chemical knowledge in the example systems or actionable knowledge for others looking to implement cooperative autonomous robots (either using the tools developed here or guidance for making a platform).

To reiterate the strengths of the paper (especially in revised form): The authors demonstrate diverse applications of autonomous use of orthogonal analytical techniques which has not yet been

demonstrated (to my knowledge), despite being a cornerstone of synthetic chemistry. The detail and availability of supplemental data and code is phenomenal. The effort to check manually the outcome of the autonomous experiments gives great confidence in both the quality of the work and the conclusion of the study, that cooperative robots and analytical instruments paired with automated liquid handling can substantially reduce the experimental burden (on humans) when exploring new chemistries.

Referee #2 (Remarks on code availability):

We briefly looked over code, it is available and well commented but cannot comment on functionality

Referee #3 (Remarks to the Author):

The authors responded to the comments one-by-one. However, the most important issue is whether there is sufficient technological progress in the field of autonomous experimental systems.

The authors are insisting that this research is autonomous by referring the concept of “automation” as “the act of making a process occur without human intervention and “autonomy” as “a paradigm where feedback and adaptive decision-making afford the system agency over the manner of its actions”. However, it is difficult to define this paper as an autonomous system because “automation” and “autonomy” can vary widely in the aspect of completeness or degree of human intervention. My major viewpoint is if the paper has a substantial technical jump compared with relating research to be published in “Nature”.

The proposed system incorporated two mobile robots, installed two analytical equipment (LC-MS, NMR) to cross-check the synthesis results, and expanded its previous function from bio to chemical synthesis by adopting commercial modules. But the mobile robots have no distinct features such as special motion planning or cooperative work. Although S/Ws are installed to operate the analytical equipment for orthogonal measurement, they don't exhibit clear progress in terms of algorithm. The authors automated the chemical synthesis, but it fell short of novelty in terms of H/W and workflow. Not even all processes are automated due to space or functional limitations.

Of course, I think this study can have its own meaning and value if it is published in more focused journals such as autonomous experiments or analysis, or in lower impact-factor journals. However, I still don't think there is enough novelty or technical progress to be published in “Nature”.

Author Rebuttals to First Revision:

Referees' comments:

Referee #1 (Remarks to the Author):

The authors have provided a thorough rebuttal to my comments and the comments of the other Referees, but unfortunately upon re-reading the manuscript, I am left with the perception that the present work is mainly a robotics and integration advance with a step backwards in terms of how automation is used to change how research is done. The simplicity of the decision making is such that a team of trained undergraduates could replicate this work in a similar amount of time by following a simple check list.

In the abstract of our original submission, we referred to robots “*that perform physical measurements and make decisions in a human-like way.*” We did not contend that the algorithm was necessarily smarter than human researchers, indeed our goal was to replicate human-level decision making across a range of chemistries. We believe that we have demonstrated this, and where there were exceptions (e.g., Extended Fig. 3), we have noted them. We do not agree that a team of undergraduates could replicate this work in the same amount of time since the algorithmic decision maker runs extremely quickly (in seconds) and the robot is capable of working 24/7. Putting timescales aside, we are not aware of any automated or autonomous chemistry studies where the work could not have been done, in principle, by trained undergraduates or postgraduates.

The goal of automated labs is not to replace researchers, but to enable new types of exploration. For this to happen, the knowledge from experiments must be meaningfully incorporated into subsequent experiments more than just knowing whether that particular experiment is worth repeating at a larger scale.

We would suggest that there is no single agreed definition of what an automated laboratory should be: there are a range of perspectives here. Nevertheless, like the Reviewer, we also do not wish to replace researchers, nor do we believe that this is possible. We do see tremendous value, though, in autonomous systems that can direct experiments into potentially interesting areas of experimental space, accepting that human interpretation and understanding will then be needed to proceed to the next set of experiments. Also, it is not quite correct to say that our approach only decides what experiments to scale up—that is one aspect, but we also demonstrate conditional diversification chemistry (first section) and autonomous host-guest bind assays (second section).

While the authors state that closed-loop operation is only currently possible in optimization, what is discovery but optimization of novelty in a large parameter space. I reject that automated curve analysis is a meaningful level of agency for a research robot of this scale.

It is unclear how one could optimize novelty in a fully closed-loop fashion for complex systems such as the supramolecular assemblies presented here. For that to be possible, we would need a quantifiable ‘novelty’ or ‘importance’ score that could be deduced from the characterization data available; that is, ¹H NMR and UPLC-MS. There are hard experimental constraints here, and some cautionary tales. For example, attempts to select novelty autonomously using narrow experimental datasets (*Nature*, **2023**, 624, 86) have been suggested to be flawed because of overreliance on a single characterisation method, which is precisely the challenge that we address here. Our systems are no less complex and they present the same pitfalls (e.g., the reactions can produce complex product mixtures). We do, however, appreciate the spirit of the reviewer’s comments, and they provide inspiration for future research. To better capture this, we have now caveated the following sentence:

“Exploratory synthesis lends itself less well to closed-loop optimisation strategies, at least in the absence of a simple quantitative ‘novelty’ or ‘importance’ metric.”

In the Conclusion section of the original manuscript, we had already stated that the autonomous decision-maker was “*less nuanced and informed*” than a human researcher, but we have now made that point even clearer by adding the following sentence:

“The level of autonomous decision making and contextual understanding is, of course, far lower than for a human researcher (Fig. 1b), but the system out-performs humans in other respects. For example, the algorithmic decisions are effectively instantaneous, providing a large acceleration over human workflows where a researcher would need to inspect all the characterisation data before proceeding further.”

We also added the following sentence to the Conclusions section:

“The inherent challenges associated with assessing ‘novelty’ or ‘importance’ of reactions in an autonomous way might suggest that we should focus instead on autonomous optimization of measurable function, such in catalysis development¹¹, but not all areas of chemistry are function led, synthetic methodology development being one example.”

I agree with Referee #2 that a journal such as Nature Synthesis is a better fit for this work although Nature Chemical Engineering could be appropriate given the focus on robotics. Either way, this work does not comprise a transformational advance but rather an incremental one.

Referee #2 (Remarks to the Author):

The authors compellingly addressed and rebutted most of the comments raised in the initial review. The acknowledgement of human input for initializing each task and fine-tuning the analysis heuristics/thresholds for each task clarifies the claim of autonomous synthetic chemistry. The additional text explaining the demonstrative purpose of two cooperative robots when only a single robot is necessary for the chosen tasks addresses a substantial apparent shortcoming in the original manuscript.

The argument that the chemistry explored is diverse and difficult to search with learning or optimization approaches is persuasive, especially regarding supramolecular assembly where there is not yet enough understanding to build predictive synthetic tools.

The main concern is that the study is a demonstration of an open-ended exploration that would have been successful no matter the outcome of the chemical studies. There is no benchmark to achieve or hypothesis to test. While the revised manuscript and rebuttal repeatedly state the project does not demonstrate solving an optimization problem, experimental design involves either hypothesis testing or some optimization challenge. Perhaps emphasis could be added on what new knowledge was gained in this study, be it chemical knowledge in the example systems or actionable knowledge for others looking to implement cooperative autonomous robots (either using the tools developed here or guidance for making a platform).

This is a helpful suggestion. We have now covered both points in the following additions to the Conclusions section:

(i) **New chemical knowledge gained from system:**

"These autonomous searches led to new chemical understanding, although that did require additional post-experiment analysis by human researchers, for example to identify the unexpected cyclization product shown in Extended Fig. 3. Likewise, in the supramolecular workflow, Supramolecular Screening 9 gave an ¹H NMR spectrum that passed the algorithm's 'hit' threshold but failed the UPLC-MS test (no matching ions observed; Supplementary Information, Scheme S32 & Fig. S121). The full nature of this structure remains undetermined, and manual attempts to produce crystals suitable for x-ray diffraction have so far failed. This illustrates how rules-based autonomous robotic searches can yield systems of potential interest, but also the challenge of characterizing them, even by hand."

(ii) Actionable knowledge for others implementing autonomous robots:

"One practical learning point for modular workflows comprising multiple concatenated software and hardware platforms is the need for very low failure rates per module. For example, the workflow illustrated here required more than a year of development and debugging before it was stable enough to carry out these experiments."

To reiterate the strengths of the paper (especially in revised form): The authors demonstrate diverse applications of autonomous use of orthogonal analytical techniques which has not yet been demonstrated (to my knowledge), despite being a cornerstone of synthetic chemistry. The detail and availability of supplemental data and code is phenomenal. The effort to check manually the outcome of the autonomous experiments gives great confidence in both the quality of the work and the conclusion of the study, that cooperative robots and analytical instruments paired with automated liquid handling can substantially reduce the experimental burden (on humans) when exploring new chemistries.

We strongly appreciate these comments; we have presented all data in detail, both good and bad, including any workflow errors, and we believe this paper gives an accurate description of the scope and limitations of this technology today.

Referee #2 (Remarks on code availability):

We briefly looked over code, it is available and well commented but cannot comment on functionality

Referee #3 (Remarks to the Author):

The authors responded to the comments one-by-one. However, the most important issue is whether there is sufficient technological progress in the field of autonomous experimental systems.

The authors are insisting that this research is autonomous by referring the concept of "automation" as "the act of making a process occur without human intervention and "autonomy" as "a paradigm where feedback and adaptive decision-making afford the system agency over the manner of its actions". However, it is difficult to define this paper as an autonomous system because "automation" and "autonomy" can vary widely in the aspect of completeness or degree of human intervention.

We used this specific definition of "automation" versus "autonomy" in the previous response to reviewers. We did not create the definition ourselves—it was taken from work of Jensen and colleagues (*Digital Discovery*, 2023, 2, 1259)—but we feel that it is a reasonable working definition. Our algorithm uses data that are obtained automatically to determine the next experimental steps across different areas of chemistry, without human input after the initial set up. We are convinced that this constitutes 'autonomy' in the common usage of this

research field. We accept that it does not represent human-like comprehension and understanding, creativity, or the ability to conceive entirely new lines of research, nor do we claim that in the paper. We do not believe that any example of this level of autonomous agency exists in the literature to date.

My major viewpoint is if the paper has a substantial technical jump compared with relating research to be published in "Nature".

The proposed system incorporated two mobile robots,

(we dealt with this point in the previous response and showed in the first revision that the workflow can be carried out with a single robot)

installed two analytical equipment (LC-MS, NMR) to cross-check the synthesis results, and expanded its previous function from bio to chemical synthesis by adopting commercial modules.

But the mobile robots have no distinct features such as special motion planning or cooperative work. Although S/Ws are installed to operate the analytical equipment for orthogonal measurement, they don't exhibit clear progress in terms of algorithm. The authors automated the chemical synthesis, but it fell short of novelty in terms of H/W and workflow. Not even all processes are automated due to space or functional limitations.

The only intervention was chemical restocking. This is a feature of all autonomous workflows to date, and that, too, could be automated in the future using mobile robots. Indeed, such a distributed approach should be more scalable (and safer) than building larger and larger installations to house all the consumables (*i.e.*, the robot could replenish consumables from a store that is located somewhere else).

Of course, I think this study can have its own meaning and value if it is published in more focused journals such as autonomous experiments or analysis, or in lower impact-factor journals. However, I still don't think there is enough novelty or technical progress to be published in "Nature".